# REPRESENTATIONAL ASPECTS OF DEPTH AND CONDITIONING IN NORMALIZING FLOWS

## ABSTRACT

Normalizing flows are among the most popular paradigms in generative modeling, especially for images, primarily because we can efficiently evaluate the likelihood of a data point. This is desirable both for evaluating the fit of a model, and for ease of training, as maximizing the likelihood can be done by gradient descent. However, training normalizing flows comes with difficulties as well: models which produce good samples typically need to be extremely deep – which comes with accompanying vanishing/exploding gradient problems. A very related problem is that they are often poorly *conditioned*: since they are parametrized as invertible maps from $\mathbb{R}^d \to \mathbb{R}^d$, and typical training data like images intuitively is lower-dimensional, the learned maps often have Jacobians that are close to being singular.

In our paper, we tackle representational aspects around depth and conditioning of normalizing flows—both for general invertible architectures, and for a particular common architecture—affine couplings.

For general invertible architectures, we prove that invertibility comes at a cost in terms of depth: we show examples where a much deeper normalizing flow model may need to be used to match the performance of a non-invertible generator.

For affine couplings, we first show that the choice of partitions isn't a likely bottleneck for depth: we show that any invertible linear map (and hence a permutation) can be simulated by a constant number of affine coupling layers, using a fixed partition. This shows that the extra flexibility conferred by 1x1 convolution layers, as in GLOW, can in principle be simulated by increasing the size by a constant factor. Next, in terms of conditioning, we show that affine couplings are universal approximators – provided the Jacobian of the model is allowed to be close to singular. We furthermore empirically explore the benefit of different kinds of padding – a common strategy for improving conditioning.

## 1 INTRODUCTION

Deep generative models are one of the lynchpins of unsupervised learning, underlying tasks spanning distribution learning, feature extraction and transfer learning. Parametric families of neural-network based models have been improved to the point of being able to model complex distributions like images of human faces. One paradigm that has received a lot attention is normalizing flows, which model distributions as pushforwards of a standard Gaussian (or other simple distribution) through an *invertible* neural network $G$. Thus, the likelihood has an explicit form via the change of variables formula using the Jacobian of $G$. Training normalizing flows is challenging due to a couple of main issues. Empirically, these models seem to require a much larger size than other generative models (e.g. GANs) and most notably, a much larger depth. This makes training challenging due to vanishing/exploding gradients. A very related problem is *conditioning*, more precisely the smallest singular value of the forward map $G$. It's intuitively clear that natural images will have a low-dimensional structure, thus a close-to-singular $G$ might be needed. On the other hand, the change-of-variables formula involves the determinant of the Jacobian of $G^{-1}$, which grows larger the more singular $G$ is.

While recently, the universal approximation power of various types of invertible architectures has been studied (Dupont et al., 2019; Huang et al., 2020) if the input is padded with a sufficiently large number of all-0 coordinates, precise quantification of the cost of invertibility in terms of the depth required and the conditioning of the model has not been fleshed out.

In this paper, we study both mathematically and empirically representational aspects of depth and conditioning in normalizing flows and answer several fundamental questions.

## 2 OVERVIEW OF RESULTS

### 2.1 RESULTS ABOUT GENERAL ARCHITECTURES

In order to guarantee that the network is invertible, normalizing flow models place significant restrictions on the architecture of the model. The most basic question we can ask is how this restriction affects the expressive power of the model — in particular, how much the depth must increase to compensate.

More precisely, we ask:

**Question 1:** is there a distribution over $\mathbb{R}^d$ which can be written as the pushforward of a Gaussian through a small, shallow generator, which cannot be approximated by the pushforward of a Gaussian through a small, shallow *layerwise invertible* neural network?

Given that there is great latitude in terms of the choice of layer architecture, while keeping the network invertible, the most general way to pose this question is to require each layer to be a function of $p$ parameters – i.e. $f = f_1 \circ f_2 \circ \cdots \circ f_\ell$ where $\circ$ denotes function composition and each $f_i : \mathbb{R}^d \to \mathbb{R}^d$ is an invertible function specified by a vector $\theta_i \in \mathbb{R}^p$ of parameters. This framing is extremely general: for instance it includes *layerwise invertible feedforward networks* in which $f_i(x) = \sigma^{\otimes d}(A_i x + b_i)$, $\sigma$ is invertible, $A_i \in \mathbb{R}^{d \times d}$ is invertible, $\theta_i = (A_i, b_i)$ and $p = d(d+1)$. It also includes popular architectures based on *affine coupling blocks* (e.g. Dinh et al. (2014; 2016); Kingma & Dhariwal (2018)) where each $f_i$ has the form $f_i(x_{S_i}, x_{[d] \setminus S_i}) = (x_{S_i}, x_{[d] \setminus S_i} \odot g_i(x_{S_i}) + h_i(x_{S_i}))$ for some $S \subset [d]$ which we revisit in more detail in the following subsection.

We answer this question in the affirmative: namely, we show for any $k$ that there is a distribution over $\mathbb{R}^d$ which can be expressed as the pushforward of a network with depth $O(1)$ and size $O(k)$ that cannot be (even very approximately) expressed as the pushforward of a Gaussian through a Lipschitz layerwise invertible network of depth smaller than $k/p$.

Towards formally stating the result, let $\theta = (\theta_1, \ldots, \theta_\ell) \in \Theta \subset \mathbb{R}^{d'}$ be the vector of all parameters (e.g. weights, biases) in the network, where $\theta_i \in \mathbb{R}^p$ are the parameters that correspond to layer $i$, and let $f_\theta : \mathbb{R}^d \to \mathbb{R}^d$ denote the resulting function. Define $R$ so that $\Theta$ is contained in the Euclidean ball of radius $R$.

We say the family $f_\theta$ is *L-Lipschitz with respect to its parameters and inputs*, if

$$\forall \theta, \theta' \in \Theta : \mathbb{E}_{x \sim \mathcal{N}(0, I_{d \times d})} \|f_\theta(x) - f_{\theta'}(x)\| \le L\|\theta - \theta'\|$$

and $\forall x, y \in \mathbb{R}^d, \|f_\theta(x) - f_\theta(y)\| \le L\|x - y\|$. [1] We will discuss the reasonable range for $L$ in terms of the weights after the Theorem statement. We show[2]:

**Theorem 1.** *For any* $k = \exp(o(d)), L = \exp(o(d)), R = \exp(o(d))$*, we have that for $d$ sufficiently large and any $\gamma > 0$ there exists a neural network $g : \mathbb{R}^{d+1} \to \mathbb{R}^d$ with $O(k)$ parameters and depth $O(1)$, s.t. for any family $\{f_\theta, \theta \in \Theta\}$ of layerwise invertible networks that are L-Lipschitz with respect to its parameters and inputs, have $p$ parameters per layer and depth at most $k/p$ we have*

$$\forall \theta \in \Theta, W_1((f_\theta)_{\#\mathcal{N}}, g_{\#\mathcal{N}}) \ge 10\gamma^2 d$$

*Furthermore, for all $\theta \in \Theta$, $KL((f_\theta)_{\#\mathcal{N}}, g_{\#\mathcal{N}}) \ge 1/10$ and $KL(g_{\#\mathcal{N}}, (f_\theta)_{\#\mathcal{N}}) \ge \frac{10\gamma^2 d}{L^2}$.*

**Remark 1:** First, note that while the number of parameters in both networks is comparable (i.e. it's $O(k)$), the invertible network is deeper, which usually is accompanied with algorithmic difficulties for training, due to vanishing and exploding gradients. For layerwise invertible generators, if we assume that the nonlinearity $\sigma$ is 1-Lipschitz and each matrix in the network has operator norm at most $M$,

---

[1]Note for architectures having trainable biases in the input layer, these two notions of Lipschitzness should be expected to behave similarly.

[2]In this Theorem and throughout, we use the standard asymptotic notation $f(d) = o(g(d))$ to indicate that $\limsup_{d \to \infty} \frac{f(d)}{g(d)} = 0$. For example, the assumption $k, L, R = \exp(o(d))$ means that for any sequence $(k_d, L_d, R_d)_{d=1}^{\infty}$ such that $\limsup_{d \to \infty} \frac{\max(\log k_d, \log L_d, \log R_d)}{d} = 0$ the result holds true.

then a depth $\ell$ network will have $L = O(M^\ell)^3$ and $p = O(d^2)$. For an affine coupling network with $g, h$ parameterized by $H$-layer networks with $p/2$ parameters each, 1-Lipschitz activations and weights bounded by $M$ as above, we would similarly have $L = O(M^{\ell H})$.

**Remark 2:** We make a couple of comments on the "hard" distribution $g$ we construct, as well as the meaning of the parameter $\gamma$ and how to interpret the various lower bounds in the different metrics. The distribution $g$ for a given $\gamma$ will in fact be close to a mixture of $k$ Gaussians, each with mean on the sphere of radius $10\gamma^2 d$ and covariance matrix $\gamma^2 I_d$. Thus this distribution has most of it's mass in a sphere of radius $O(\gamma^2 d)$ — so the Wasserstein guarantee gives close to a trivial approximation for $g$. The KL divergence bounds are derived by so-called transport inequalities between KL and Wasserstein for subgaussian distributions Bobkov & Götze (1999). The discrepancy between the two KL divergences comes from the fact that the functions $g, f_\theta$ may have different Lipschitz constants, hence the tails of $g_{\#\mathcal{N}}$ and $f_{\#\mathcal{N}}$ behave differently. In fact, if the function $f_\theta$ had the same Lipschitz constant as $g$, both KL lower bounds would be on the order of a constant.

## 2.2 RESULTS ABOUT AFFINE COUPLING ARCHITECTURES

Next, we prove several results for a particularly common normalizing flow architectures: those based on affine coupling layers (Dinh et al., 2014; 2016; Kingma & Dhariwal, 2018). The appeal of these architecture comes from training efficiency. Although layerwise invertible neural networks (i.e. networks for which each layer consists of an invertible matrix and invertible pointwise nonlinearity) seem like a natural choice, in practice these models have several disadvantages: for example, computing the determinant of the Jacobian is expensive unless the weight matrices are restricted.

Consequently, it's typical for the transformations in a flow network to be constrained in a manner that allows for efficient computation of the Jacobian determinant. The most common building block is an *affine coupling* block, originally proposed by Dinh et al. (2014; 2016). A coupling block partitions the coordinates $[d]$ into two parts: $S$ and $[d] \setminus S$, for a subset $S$ with $|S|$ containing around half the coordinates of $d$. The transformation then has the form:

**Definition 1.** An *affine coupling block* is a map $f : \mathbb{R}^d \to \mathbb{R}^d$, s.t. $f(x_S, x_{[d]\setminus S}) = (x_S, x_{[d]\setminus S} \odot s(x_S) + t(x_S))$

Of course, the modeling power will be severely constrained if the coordinates in $S$ never change: so typically, flow models either change the set $S$ in a fixed or learned way (e.g. alternating between different partitions of the channel in Dinh et al. (2016) or applying a learned permutation in Kingma & Dhariwal (2018)). Of course, a permutation is a discrete object, so difficult to learn in a differentiable manner – so Kingma & Dhariwal (2018) simply learns an invertible linear function (i.e. a 1x1 convolution) as a differentiation-friendly relaxation thereof.

### 2.2.1 THE EFFECT OF CHOICE OF PARTITION ON DEPTH

The first question about affine couplings we ask is how much of a saving in terms of the depth of the network can one hope to gain from using learned partitions (ala GLOW) as compared to a fixed partition. More precisely:

**Question 2:** Can models like Glow (Kingma & Dhariwal, 2018) be simulated by a sequence of affine blocks with a fixed partition without increasing the depth by much?

We answer this question in the affirmative at least for equally sized partitions (which is what is typically used in practice). We show the following surprising fact: consider an arbitrary partition $(S, [2d] \setminus S)$ of $[2d]$, such that $S$ satisfies $|S| = d$, for $d \in \mathbb{N}$. Then for any invertible matrix $T \in \mathbb{R}^{2d \times 2d}$, the linear map $T : \mathbb{R}^{2d} \to \mathbb{R}^{2d}$ can be exactly represented by a composition of $O(1)$ affine coupling layers that are *linear*, namely have the form $L_i(x_S, x_{[2d]\setminus S}) = (x_S, B_i x_{[2d]\setminus S} + A_i x_S)$ or $L_i(x_S, x_{[2d]\setminus S}) = (C_i x_S + D_i x_{[2d]\setminus S}, x_{[2d]\setminus S})$ for matrices $A_i, B_i, C_i, D_i \in \mathbb{R}^{d \times d}$, s.t. each $B_i, C_i$ is diagonal. For convenience of notation, without loss of generality let $S = [d]$. Then, each of the layers $L_i$ is a matrix of the form $\begin{bmatrix} I & 0 \\ A_i & B_i \end{bmatrix}$ or $\begin{bmatrix} C_i & D_i \\ 0 & I \end{bmatrix}$, where the rows and columns are partitioned into blocks of size $d$.

With this notation in place, we show the following theorem:

---

[3]Note, our theorem applies to exponentially large Lipschitz constants.

**Theorem 2.** *For all $d \geq 4$, there exists a $k \leq 24$ such that for any invertible $T \in \mathbb{R}^{2d \times 2d}$ with $\det(T) > 0$, there exist matrices $A_i, D_i \in \mathbb{R}^{d \times d}$ and diagonal matrices $B_i, C_i \in \mathbb{R}_{\geq 0}^{d \times d}$ for all $i \in [k]$ such that*

$$T = \prod_{i=1}^{k} \begin{bmatrix} I & 0 \\ A_i & B_i \end{bmatrix} \begin{bmatrix} C_i & D_i \\ 0 & I \end{bmatrix}$$

Note that the condition $\det(T) > 0$ is required, since affine coupling networks are always orientation-preserving. Adding one diagonal layer with negative signs suffices to model general matrices. In particular, since permutation matrices are invertible, this means that any applications of permutations to achieve a different partition of the inputs (e.g. like in Glow (Kingma & Dhariwal, 2018)) can in principle be represented as a composition of not-too-many affine coupling layers, indicating that the flexibility in the choice of partition is not the representational bottleneck.

It's a reasonable to ask how optimal the $k \leq 24$ bound is – we supplement our upper bound with a lower bound, namely that $k \geq 3$. This is surprising, as naive parameter counting would suggest $k = 2$ might work. Namely, we show:

**Theorem 3.** *For all $d \geq 4$ and $k \leq 2$, there exists an invertible $T \in \mathbb{R}^{2d \times 2d}$ with $\det(T) > 0$, s.t. for all $A_i, D_i \in \mathbb{R}^{d \times d}$ and for all diagonal matrices $B_i, C_i \in \mathbb{R}_{\geq 0}^{d \times d}, i \in [k]$ it holds that*

$$T \neq \prod_{i=1}^{k} \begin{bmatrix} I & 0 \\ A_i & B_i \end{bmatrix} \begin{bmatrix} C_i & D_i \\ 0 & I \end{bmatrix}$$

Beyond the relevance of this result in the context of how important the choice of partitions is, it also shows a lower bound on the depth for an equal number of *nonlinear* affine coupling layers (even with quite complex functions $s$ and $t$ in each layer) – since a nonlinear network can always be linearized about a (smooth) point to give a linear network with the same number of layers. In other words, studying linear affine coupling networks lets us prove a *depth lower bound/depth separation* for nonlinear networks for free.

Finally, in Section 5.3, we include an empirical investigation of our theoretical results on synthetic data, by fitting random linear functions of varying dimensionality with linear affine networks of varying depths in order to see the required number of layers. The results there suggest that the constant in the upper bound is quite loose – and the correct value for $k$ is likely closer to the lower bound – at least for random matrices.

### 2.2.2 UNIVERSAL APPROXIMATION WITH ILL-CONDITIONED AFFINE COUPLING NETWORKS

Finally, we turn to universal approximation and the close ties to conditioning. Namely, a recent work (Theorem 1 of Huang et al. (2020)) showed that deep affine coupling networks are universal approximators if we allow the training data to be padded with sufficiently many zeros. While zero padding is convenient for their analysis (in fact, similar proofs have appeared for other invertible architectures like Augmented Neural ODEs (Zhang et al.)), in practice models trained on zero-padded data often perform poorly (see Appendix C).

In fact, we show that neither padding nor depth is necessary representationally: shallow models without zero padding are already universal approximators in Wasserstein.

**Theorem 4** (Universal approximation without padding). *Suppose that $P$ is the standard Gaussian measure in $\mathbb{R}^n$ with $n$ even and $Q$ is a distribution on $\mathbb{R}^n$ with bounded support and absolutely continuous with respect to the Lebesgue measure. Then for any $\epsilon > 0$, there exists a depth-3 affine coupling network $g$, with maps $s, t$ represented by feedforward ReLU networks such that $W_2(g_{\#}P, Q) \leq \epsilon$.*

**Remark 1:** A shared caveat of the universality construction in Theorem 4 with the construction in Huang et al. (2020) is that the resulting network is poorly conditioned. In the case of the construction in Huang et al. (2020), this is obvious because they pad the $d$-dimensional training data with $d$ additional zeros, and a network that takes as input a Gaussian distribution in $\mathbb{R}^{2d}$ (i.e. has full support) and outputs data on $d$-dimensional manifold (the space of zero padded data) must have a singular

Jacobian almost everywhere.[4] In the case of Theorem 4, the condition number of the network blows up at least as quickly as $1/\epsilon$ as we take the approximation error $\epsilon \to 0$, so this network is also ill-conditioned if we are aiming for a very accurate approximation.

**Remark 2:** Based on Theorem 3, the condition number blowup of either the Jacobian or the Hessian is necessary for a shallow model to be universal, even when approximating well-conditioned linear maps (see Remark 7). The network constructed in Theorem 4 is also consistent with the lower bound from Theorem 1, because the network we construct in Theorem 4 is highly non-Lipschitz and uses many parameters per layer.

## 3 RELATED WORK

On the empirical side, flow models were first popularized by Dinh et al. (2014), who introduce the NICE model and the idea of parametrizing a distribution as a sequence of transformations with triangular Jacobians, so that maximum likelihood training is tractable. Quickly thereafter, Dinh et al. (2016) improved the affine coupling block architecture they introduced to allow non-volume-preserving (NVP) transformations, Papamakarios et al. (2017) introduced an autoregressive version, and finally Kingma & Dhariwal (2018) introduced 1x1 convolutions in the architecture, which they view as relaxations of permutation matrices—intuitively, allowing learned partitions for the affine blocks. Subsequently, there have been variants on these ideas: (Grathwohl et al., 2018; Dupont et al., 2019; Behrmann et al., 2018) viewed these models as discretizations of ODEs and introduced ways to approximate determinants of non-triangular Jacobians, though these models still don't scale beyond datasets the size of CIFAR10. The conditioning/invertibility of trained models was experimentally studied in (Behrmann et al., 2019), along with some "adversarial vulnerabilities" of the conditioning. Mathematically understanding the relative representational power and statistical/algorithmic implications thereof for different types of generative models is still however a very poorly understood and nascent area of study.

Most closely related to our results are the recent works of Huang et al. (2020) and Zhang et al.. Both prove universal approximation results for invertible architectures (the former affine couplings, the latter neural ODEs) if the input is allowed to be padded with zeroes. As already expounded upon in the previous sections – our results prove universal approximation even without padding, but we focus on more fine-grained implications to depth and conditioning of the learned model. Another work (Kong & Chaudhuri, 2020) studies the representational power of Sylvester and Householder flows, normalizing flow architectures which are quite different from affine coupling networks. In particular, they prove a depth lower bound for local planar flows with bounded weights; for planar flows, our general Theorem 1 can also be applied, but the resulting lower bound instances are very different (ours targets multimodality, theirs targets tail behavior).

More generally, there are various classical results that show a particular family of generative models can closely approximate most sufficiently regular distributions over some domain. Some examples are standard results for mixture models with very mild conditions on the component distribution (e.g. Gaussians, see (Everitt, 2014)); Restricted Boltzmann Machines and Deep Belief Networks (Montúfar et al., 2011; Montufar & Ay, 2011); GANs (Bailey & Telgarsky, 2018).

## 4 PROOF SKETCH OF THEOREM 1: DEPTH LOWER BOUNDS ON INVERTIBLE MODELS

In this section we sketch the proof of Theorem 1. The intuition behind the $k/p$ bound on the depth relies on parameter counting: a depth $k/p$ invertible network will have $k$ parameters in total ($p$ per layer)—which is the size of the network we are trying to represent. Of course, the difficulty is that we need more than $f_\theta, g$ simply not being identical: we need a quantitative bound in various probability metrics.

The proof will proceed as follows. First, we will exhibit a large family of distributions (of size $\exp(kd)$), s.t. each pair of these distributions has a large pairwise Wasserstein distance between them. Moreover, each distribution in this family will be approximately expressible as the pushforward of

---

[4]Alternatively, we could feed a degenerate Gaussian supported on a $d$-dimensional subspace into the network as input, but there is no way to train such a model using maximum-likelihood training, since the prior is degenerate.

the Gaussian through a small neural network. Since the family of distributions will have a large pairwise Wasserstein distance, by the triangle inequality, no other distribution can be close to two distinct members of the family.

Second, we can count the number of "approximately distinct" invertible networks of depth $l$: each layer is described by $p$ weights, hence there are $lp$ parameters in total. The Lipschitzness of the neural network in terms of its parameters then allows to argue about discretizations of the weights.

Formally, we show the following lemma:

**Lemma 1** (Large family of well-separated distributions). *For every $k = o(exp(d))$, for $d$ sufficiently large and $\gamma > 0$ there exists a family $\mathcal{D}$ of distributions, s.t. $|\mathcal{D}| \geq \exp(kd/20)$ and:*

1. *Each distribution $p \in \mathcal{D}$ is a mixture of $k$ Gaussians with means $\{\mu_i\}_{i=1}^k, \|\mu_i\|^2 = 20\gamma^2 d$ and covariance $\gamma^2 I_d$.*

2. *$\forall p \in \mathcal{D}$ and $\forall \epsilon > 0$, we have $W_1(p, g_{\#\mathcal{N}}) \leq \epsilon$ for a neural network $g$ with at most $O(k)$ parameters.[5]*

3. *For any $p, p' \in \mathcal{D}$, $W_1(p, p') \geq 20\gamma^2 d$.*

The proof of this lemma will rely on two ideas: first, we will show that there is a family of distributions consisting of mixtures of Gaussians with $k$ components – s.t. each pair of members of this family is far in $W_1$ distance, and each member in the family can be approximated by the pushforward of a network of size $O(k)$.

The reason for choosing mixtures is that it's easy to lower bound the Wasserstein distance between two mixtures with equal weights and covariance matrices in terms of the distances between the means. We show this as Lemma 5 in Appendix A.

Given this, to design a family of mixtures of Gaussians with large pairwise Wasserstein distance, it suffices to construct a large family of $k$-tuples for the means, s.t. for each pair of $k$-tuples $(\{\mu_i\}_{i=1}^k, \{\nu_i\}_{i=1}^k)$, there exists a set $S \subseteq [k], |S| \geq k/10$, s.t. $\forall i \in S, \min_{1 \leq j \leq k} \|\mu_i - \nu_j\|^2 \geq 20\gamma^2 d$. We do this by leveraging ideas from coding theory (the Gilbert-Varshamov bound Gilbert (1952); Varshamov (1957)). Namely, we first pick a set of $\exp(\Omega(d))$ vectors of norm $20\gamma^2 d$, each pair of which has a large distance; second, we pick a large number ($\exp(\Omega(kd))$) of $k$-tuples from this set at random, and show with high probability, no pair of tuples intersect in more than $k/10$ elements. This is subsumed by Lemmas 6 and 7 in Section A.

To handle part 2 of Lemma 1, we also show that a mixture of $k$ Gaussians can be approximated as the pushforward of a Gaussian through a network of size $O(k)$. The idea is rather simple: the network will use a sample from a standard Gaussian in $\mathbb{R}^{d+1}$. We will subsequently use the first coordinate to implement a "mask" that most of the time masks all but one randomly chosen coordinate in $[k]$. The remaining coordinates are used to produce a sample from each of the components in the Gaussian, and the mask is used to select only one of them. For details, see Section A.

With this lemma in hand, we finish the Wasserstein lower bound with a standard epsilon-net argument, using the parameter Lipschitzness of the invertible networks by showing the number of "different" invertible neural networks is on the order of $O\left((LR)^{d'}\right)$. This is Lemma 8 in Appendix A. The proof of Theorem 1 can then be finished by triangle inequality: since the family of distributions has large Wasserstein distance, by the triangle inequality, no other distribution can be close to two distinct members of the family. Finally, KL divergence bounds can be derived from the Bobkov-Götze inequality Bobkov & Götze (1999), which lower bounds KL divergence by the squared Wasserstein distance. The details are in Section A.

## 5    PROOF SKETCH OF THEOREMS 2 AND 3: SIMULATING LINEAR FUNCTIONS WITH AFFINE COUPLINGS

In this section, we will prove Theorems 3 and 2. Before proceeding to the proofs, we will introduce a bit of helpful notation. We let $GL^+(2d, \mathbb{R})$ denote the group of $2d \times 2d$ matrices with positive

---

[5]The size of $g$ doesn't indeed depend on $\epsilon$. The weights in the networks will simply grow as $\epsilon$ becomes small.

determinant (see Artin (2011) for a reference on group theory). The lower triangular linear affine coupling layers are the subgroup $\mathcal{A}_{\mathcal{L}} \subset GL^+(2d, \mathbb{R})$ of the form

$$\mathcal{A}_{\mathcal{L}} = \left\{ \begin{bmatrix} I & 0 \\ A & B \end{bmatrix} : A \in \mathbb{R}^{d \times d}, B \text{ is diagonal with positive entries} \right\},$$

and likewise the upper triangular linear affine coupling layers are the subgroup $\mathcal{A}_{\mathcal{U}} \subset GL^+(2d, \mathbb{R})$ of the form

$$\mathcal{A}_{\mathcal{U}} = \left\{ \begin{bmatrix} C & D \\ 0 & I \end{bmatrix} : D \in \mathbb{R}^{d \times d}, C \text{ is diagonal with positive entries} \right\}.$$

Finally, define $\mathcal{A} = \mathcal{A}_{\mathcal{L}} \cup \mathcal{A}_{\mathcal{U}} \subset GL^+(2d, \mathbb{R})$. This set is not a subgroup because it is not closed under multiplication. Let $\mathcal{A}^k$ denote the $k$th power of $\mathcal{A}$, i.e. all elements of the form $a_1 \cdots a_k$ for $a_i \in \mathcal{A}$.

## 5.1 Upper Bound

The main result of this section is the following:

**Theorem 5** (Restatement of Theorem 2). *There exists an absolute constant $1 < K \leq 47$ such that for any $d \geq 1$, $GL^+(2d, \mathbb{R}) = \mathcal{A}^K$.*

In other words, any linear map with positive determinant ("orientation-preserving") can be implemented using a bounded number of linear affine coupling layers. Note that there is a difference in a factor of two between the counting of layers in the statement of Theorem 2 and the counting of matrices in Theorem 5, because each layer is composed of two matrices.

In group-theoretic language, this says that $\mathcal{A}$ generates $GL^+(2d, \mathbb{R})$ and furthermore the diameter of the corresponding (uncountably infinite) Cayley graph is upper bounded by a constant independent of $d$. The proof relies on the following two structural results. The first one is about representing permutation matrices, up to sign, using a constant number of linear affine coupling layers:

**Lemma 2.** *For any permutation matrix $P \in \mathbb{R}^{2d \times 2d}$, there exists $\tilde{P} \in \mathcal{A}^{21}$ with $|\tilde{P}_{ij}| = |P_{ij}|$ for all $i, j$.*

The second one proves how to represent using a constant number of linear affine couplings matrices with special eigenvalue structure:

**Lemma 3.** *Let $M$ be an arbitrary invertible $d \times d$ matrix with distinct real eigenvalues and $S$ be a $d \times d$ lower triangular matrix with the same eigenvalues as $M^{-1}$. Then $\begin{bmatrix} M & 0 \\ 0 & S \end{bmatrix} \in \mathcal{A}^4$.*

Given these Lemmas, we briefly describe the strategy to prove Theorem 5. Every matrix has a an $LUP$ factorization Horn & Johnson (2012) into a lower-triangular, upper-triangular, and permutation matrix. Lemma 2 takes care of the permutation part, so what remains is building an arbitrary lower/upper triangular matrix; because the eigenvalues of lower-triangular matrices are explicit, a careful argument allows us to reduce this to Lemma 3. All the proofs are in Section B.

## 5.2 Lower Bound

We proceed to the lower bound. Note, a simple parameter counting argument shows that for sufficiently large $d$, at least four affine coupling layers are needed to implement an arbitrary linear map (each affine coupling layer has only $d^2 + d$ parameters whereas $GL_+(2d, \mathbb{R})$ is a Lie group of dimension $4d^2$). Perhaps surprisingly, it turns out that four affine coupling layers *do not* suffice to construct an arbitrary linear map. We prove this in the following Theorem.

**Theorem 6** (Restatement of Theorem 3). *For $d \geq 4$, $\mathcal{A}^4$ is a proper subset of $GL_+(2d, \mathbb{R})$. In other words, there exists a matrix $T \in GL_+(2d, \mathbb{R})$ which is not in $\mathcal{A}^4$.*

Again, this translates to the result in Theorem 3 because each layer corresponds to two matrices — so this shows two layers are not enough to get arbitrary matrices. The key observation is that matrices in $\mathcal{A}_{\mathcal{L}} \mathcal{A}_{\mathcal{U}} \mathcal{A}_{\mathcal{L}} \mathcal{A}_{\mathcal{U}}$ satisfy a strong algebraic invariant which is not true of arbitrary matrices. This invariant can be expressed in terms of the *Schur complement* Zhang (2006):

**Lemma 4.** *Suppose that $T = \begin{bmatrix} X & Y \\ Z & W \end{bmatrix}$ is an invertible $2d \times 2d$ matrix and suppose there exist matrices $A, E \in \mathbb{R}^{d \times d}, D, H \in \mathbb{R}^{d \times d}$ and diagonal matrices $B, F \in \mathbb{R}^{d \times d}, C, G \in \mathbb{R}^{d \times d}$ such that*

$$T = \begin{bmatrix} I & 0 \\ A & B \end{bmatrix} \begin{bmatrix} C & D \\ 0 & I \end{bmatrix} \begin{bmatrix} I & 0 \\ E & F \end{bmatrix} \begin{bmatrix} G & H \\ 0 & I \end{bmatrix}.$$

*Then the Schur complement $T/X := W - ZX^{-1}Y$ is similar to $X^{-1}C$: more precisely, if $U = Z - AX$ then $T/X = UX^{-1}CU^{-1}$.*

The proof of this Lemma is presented in Appendix B, as well as the resulting proof of Theorem 6. We remark that the argument in the proof is actually fairly general; it can be shown, for example, that for a random choice of $X$ and $W$ from the Ginibre ensemble, that $T$ cannot typically be expressed in $\mathcal{A}_4$. So there are significant restrictions on what matrices can be expressed even four affine coupling layers.

**Remark 7** (Connection to Universal Approximation)**.** As mentioned earlier, this lower bound shows that the map computed by general 4-layer affine coupling networks is quite restricted in its local behavior (it's Jacobian cannot be arbitrary). This implies that smooth 4-layer affine coupling networks, where smooth means the Hessian (of each coordinate of the output) is bounded in spectral norm, cannot be universal function approximators as they cannot even approximate some linear maps. In contrast, if we allow the computed function to be very jagged then three layers are universal (see Theorem 4).

### 5.3 Experimental results

We also verify the bounds from this section. At least on randomly chosen matrices, the correct bound is closer to the lower bound. Precisely, we generate (synthetic) training data of the form $Az$, where $z \sim \mathcal{N}(0, I)$ for a fixed $d \times d$ square matrix $A$ with random standard Gaussian entries and train a linear affine coupling network with $n = 1, 2, 4, 8, 16$ layers by minimizing the loss $\mathbb{E}_{z \sim \mathcal{N}(0,I)} \left[ (f_n(z) - Az)^2 \right]$. We are training this "supervised" regression loss instead of the standard unsupervised likelihood loss to minimize algorithmic (training) effects as the theorems are focusing on the representational aspects. The results for $d = 16$ are shown in Figure 1, and more details are in Section C. To test a different distribution other than the Gaussian ensemble, we also generated random Toeplitz matrices with constant diagonals by sampling the value for each diagonal from a standard Gaussian and performed the same regression experiments. We found the same dependence on number of layers but an overall higher error, suggesting that that this distribution is slightly 'harder'. We provide results in Section C. We also regress a nonlinear RealNVP architecture on the same problems and see a similar increase in representational power though the nonlinear models seem to require more training to reach good performance.

**Additional Remarks** Finally, we also note that there are some surprisingly simple functions that cannot be *exactly* implemented by a finite affine coupling network. For instance, an entrywise $\tanh$ function (i.e. an entrywise nonlinearity) cannot be exactly represented by any finite affine coupling network, regardless of the nonlinearity used. Details of this are in Appendix E.

## 6 Proof Sketch of Theorem 4: Universal Approximation with Ill-Conditioned Affine Coupling Networks

In this section, we sketch the proof of Theorem 4 to show how to approximate a distribution in $\mathbb{R}^n$ using three layers of affine coupling networks, where the dimension $n = 2d$ is even. The partition in the affine coupling network is between the first $d$ coordinates and second $d$ coordinates in $\mathbb{R}^{2d}$.

The first element in the proof is a well-known theorem from optimal transport called Brenier's theorem, which states that for $Q$ a probability measure over $\mathbb{R}^n$ satisfying weak regularity conditions (see Theorem 9 in Section D), there exists a map $\varphi : \mathbb{R}^n \to \mathbb{R}^n$ such that if $X \sim N(0, I_{n \times n})$, then the pushforward $\varphi_\#(X)$ is distributed according to $Q$.

The proof then proceeds by using a lattice-based encoding and decoding scheme. Concretely, let $\epsilon > 0$ be a small constant, to be taking sufficiently small. Let $\epsilon' \in (0, \epsilon)$ be a further constant, taken sufficiently small with respect to $\epsilon$ and similar for $\epsilon''$ wrt $\epsilon'$. Let the input to the affine coupling

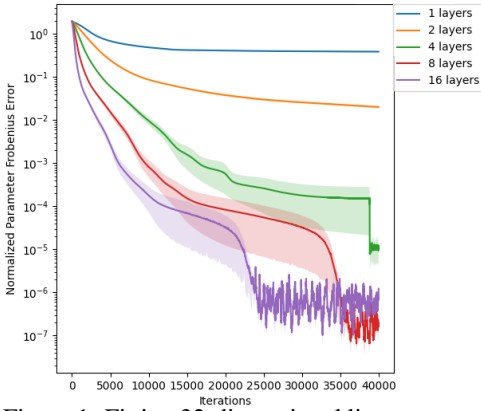

Figure 1: Fitting 32-dimensional linear maps on a using $n$-layer linear affine coupling networks. The squared Frobenius error is normalized by $1/d^2$ so it is independent of dimensionality. We shade the standard error regions of these losses across the seeds tried.

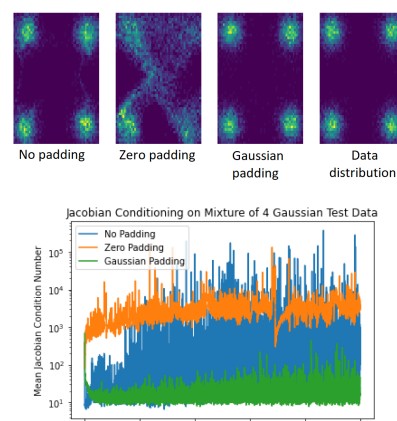

Figure 2: Fitting a 4-component mixture of Gaussians using a RealNVP model with no padding, zero padding and Gaussian padding.

network be $X = (X_1, X_2)$ such that $X_1 \sim N(0, I_{d \times d})$ and $X_2 \sim N(0, I_{d \times d})$. Let $f(x)$ be the map which rounds $x \in \mathbb{R}^d$ to the closest grid point in the lattice $\epsilon \mathbb{Z}^d$ and define $g(x) = x - f(x)$. Note that for a point of the form $z = f(x) + \epsilon' y$ for $y$ which is not too large, we have that $f(z) = f(x)$ and $g(z) = y$. Suppose the optimal transportation map from Brenier's Theorem is $\varphi(x) = (\varphi_1(x), \varphi_2(x))$ where $\varphi_1, \varphi_2 : \mathbb{R}^d \to \mathbb{R}^n$ correspond to the two halves of the output. Now we consider the following sequence of maps, all which form an affine coupling layer:

$$(X_1, X_2) \mapsto (X_1, \epsilon' X_2 + f(X_1)) \tag{1}$$
$$\mapsto (f(\varphi_1(f(X_1), X_2)) + \epsilon' \varphi_2(f(X_1), X_2) + O(\epsilon''), \epsilon' X_2 + f(X_1)) \tag{2}$$
$$\mapsto (f(\varphi_1(f(X_1), X_2)) + \epsilon' \varphi_2(f(X_1), X_2) + O(\epsilon''), \varphi_2(f(X_1), X_2) + O(\epsilon''/\epsilon')). \tag{3}$$

To explicitly see why the above are affine coupling layers, in the first step we take $s_1(x) = \log(\epsilon') \vec{1}$ and $t_1(x) = f(x)$. In the second step, we take $s_2(x) = \log(\epsilon'') \vec{1}$ and $t_2$ is defined by $t_2(x) = f(\varphi_1(f(x), g(x))) + \epsilon' \varphi_2(f(x), g(x))$. In the third step, we take $s_3(x) = \log(\epsilon'') \vec{1}$ and define $t_3(x) = \frac{g(x)}{\epsilon'}$. Taking sufficiently good approximations to all of the maps allows to approximate this map with neural networks, which we formalize in Appendix D.

## 6.1 EXPERIMENTAL RESULTS

On the empirical side, we explore the effect that different types of padding has on the training on various synthetic datasets. For Gaussian padding, this means we add to the $d$-dimensional training data point, an additional $d$ dimensions sampled from $N(0, I_d)$. We consistently observe that zero padding has the worst performance and Gaussian padding has the best performance. On Figure 2 we show the performance of a simple RealNVP architecture trained via max-likelihood on a mixture of 4 Gaussians, as well as plot the condition number of the Jacobian during training for each padding method. The latter gives support to the fact that conditioning is a major culprit for why zero padding performs so badly. In Appendix C.2 we provide figures from more synthetic datasets.

## 7 CONCLUSION

Normalizing flows are one of the most heavily used generative models across various domains, though we still have a relatively narrow understanding of their relative pros and cons compared to other models. In this paper, we tackled representational aspects of two issues that are frequent sources of training difficulties, depth and conditioning. We hope this work will inspire more theoretical study of fine-grained properties of different generative models.

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

Han Zhang, Xi Gao, Jacob Unterman, and Tom Arodz. Approximation capabilities of neural odes and invertible residual networks.

## A   MISSING PROOFS FOR SECTION 4

### A.1   WASSERSTEIN DISTANCE FOR MIXTURES

**Lemma 5.** *Let $\mu$ and $\nu$ be two mixtures of $k$ spherical Gaussians in $d$ dimensions with mixing weights $1/k$, means $(\mu_1, \mu_2, \ldots, \mu_k)$ and $(\nu_1, \nu_2, \ldots, \nu_k)$ respectively, and with all of the Gaussians having spherical covariance matrix $\gamma^2 I$ for some $\gamma > 0$. Suppose that there exists a set $S \subseteq [k]$ with $|S| \geq k/10$ such that for every $i \in S$,*

$$\min_{1 \leq j \leq k} \|\mu_i - \nu_j\|^2 \geq 20\gamma^2 d.$$

*Then $W_1(\mu, \nu) = \Omega(\gamma\sqrt{d})$.*

*Proof.* By the dual formulation of Wasserstein distance (Kantorovich-Rubinstein Theorem) Villani (2003), we have $W_1(\mu, \nu) = \sup_\varphi \left[ \int \varphi d\mu - \int \varphi d\nu \right]$ where the supremum is taken over all 1-Lipschitz functions $\varphi$. Towards lower bounding this, consider $\varphi(x) = \max(0, 2\gamma\sqrt{d} - \min_{i \in S} \|x_i - \mu_i\|)$ and note that this function is 1-Lipschitz and always valued in $[0, 2\gamma\sqrt{d}]$. For a single Gaussian $Z \sim \mathcal{N}(0, \gamma^2 I_{d \times d})$, observe that

$$\mathbb{E}_{Z \sim \mathcal{N}(0,\gamma^2 I)}[\max(0, 2\gamma\sqrt{d} - \|Z\|)] \geq 2\gamma\sqrt{d} - \mathbb{E}_{Z)}[\|Z\|] \geq 2\gamma\sqrt{d} - \sqrt{\mathbb{E}_{Z \sim \mathcal{N}}[\|Z\|^2]} \geq \gamma\sqrt{d}.$$

Therefore, we see that $\int \varphi d\mu = \Omega(\gamma\sqrt{d})$ by combining the above calculation with the fact that at least $1/10$ of the centers for $\mu$ are in $S$. On the other hand, for $Z \sim \mathcal{N}(0, \gamma^2 I_{d \times d})$ we have

$$\Pr(\|Z\|^2 \geq 10\gamma^2 d) \leq 2e^{-10d}$$

(e.g. by Bernstein's inequality Vershynin (2018), as $\|Z\|^2$ is a sum of squares of Gaussians, i.e. a $\chi^2$-random variable). In particular, since the points in $S$ do not have a close point in $\{\nu_i\}_{i=1}^k$, we similarly have $\int \varphi d\nu = O(e^{-10d}\gamma\sqrt{d}) = o(\gamma\sqrt{d})$, since very little mass from each Gaussian in $\nu_i$ lands in the support of $\varphi$ by the separation assumption. Combining the bounds gives the result.   $\square$

### A.2   CONSTRUCTING TUPLES OF WELL-SEPARATED MEANS

First, by elementary Chernoff bounds, we have the following result:

**Lemma 6** (Large family of well-separated points)**.** *Let $\epsilon > 0$. There exists a set $\{v_1, v_2, \ldots, v_N\}$ of vectors $v_i \in \mathbb{R}^d$, $\|v_i\| = 1$ with $N = \exp(d\epsilon^2/4)$, s.t. $\|v_i - v_j\|^2 \geq 2(1 - \epsilon)$ for all $i \neq j$.*

*Proof.* Recall that for a random unit vector $v$ on the sphere in $d$ dimensions, $\Pr(v_i > t/\sqrt{d}) \leq e^{-t^2/2}$. (This is a basic fact about spherical caps, see e.g. Rao (2011)). By spherical symmetry and the union bound, this means for two unit vectors $v, w$ sampled uniformly at random $\Pr(|\langle v, w \rangle| > t/\sqrt{d}) \leq 2e^{-t^2/2}$. Taking $t = \epsilon\sqrt{d}$ gives that the probability is $2e^{-d\epsilon^2/2}$; therefore if draw $N$ i.i.d. vectors, the probability that two have inner product larger than $\epsilon$ in absolute value is at most $N^2 e^{-d\epsilon^2/2} < 1$ if $N = e^{d\epsilon^2/4}$, which in particular implies such a collection of vectors exists.   $\square$

To construct tuples with small intersections, we use the following result by Rödl:

**Lemma 7** (Rödl & Thoma (1996))**.** *There exists a set consisting of $(\frac{N}{2k})^{k/10}$ subsets of size $k$ of $[N]$, s.t. no pair of subsets intersect in more than $k/10$ elements.*

### A.3   EPSILON-NET COUNT

The following lemma is immediate:

**Lemma 8.** *Suppose that $\Theta \subset \mathbb{R}^{d'}$ is contained in a ball of radius $R > 0$ and $f_\theta$ is a family of invertible layerwise networks which is $L$-Lipschitz with respect to its parameters. Then there exists a set of neural networks $S_\epsilon = \{f_i\}$, s.t. $|S_\epsilon| = O\left(\left(\frac{LR}{\epsilon}\right)^{d'}\right)$ and for every $\theta \in \Theta$ there exists a $f_i \in S_\epsilon$, s.t. $\mathbb{E}_{x \sim N(0, I_{d \times d})}\|f_\theta(x) - f_i(x)\|_\infty \leq \epsilon$.*

A.4    SIMULATING A MIXTURE WITH A NEURAL NETWORK

**Lemma 9.** *Let $p : \mathbb{R}^d \to \mathbb{R}^+$ be a mixture of $k$ Gaussians with means $\{\mu_i\}_{i=1}^k$, $\|\mu_i\|^2 = 20\gamma^2 d$ and covariance $\gamma^2 I_d$. Then, $\forall \epsilon > 0$, we have $W_1(p, g_{\#\mathcal{N}}) \le \epsilon$ for a neural network $g$ with $O(k)$ parameters.*[6]
*Moreover, for every 1-Lipschitz $\phi : \mathbb{R}^d \to \mathbb{R}^+$ and $X \sim g_{\#\mathcal{N}}$, $\phi(X)$ is $O(\gamma^2 d)$-subgaussian.*

*Proof.* We will use a construction similar to Arora et al. (2017). Since the latent variable dimension is $d + 1$, the idea is to use the first variable, say $h$ as input to a "selector" circuit which picks one of the components of the mixture with approximately the right probability, then use the remaning dimensions—say variable $z$, to output a sample from the appropriate component.

For notational convenience, let $M = \sqrt{20\gamma^2 d}$. Let $\{h_i\}_{i=1}^{k-1}$ be real values that partition $\mathbb{R}$ into $k$ intervals that have equal probability under the Gaussian measure. Then, the map

$$\tilde{f}(h, z) = \gamma z + \sum_{i=1}^k \mathbb{1}(h \in (h_{i-1}, h_i]) \mu_i \tag{4}$$

exactly generates the desired mixture, where $h_0$ is understood to be $-\infty$ and $h_k = +\infty$.

To construct $g$, first we approximate the indicators using two ReLUs, s.t. we design for each interval $(h_{i-1}, h_i]$ a function $\tilde{1}_i$, s.t.:
(1) $\tilde{1}_i(h) = \mathbb{1}(h \in (h_{i-1}, h_i])$ unless $h \in [h_{i-1}, h_{i-1} + \delta_{i-1}^+] \cup [h_i - \delta_i^-, h_i]$, and the Gaussian measure of the union of the above two intervals is $\delta$.
(2) $\sum_i \tilde{1}_i(h) = 1$.
The constructions of the functions $\tilde{1}_i$ above can be found in Arora et al. (2017), Lemma 3. We subsequently construct the neural network $f(h, z)$ using ReLUs defined as

$$f(h, z) = \gamma z + \sum_{i=1}^k \left( \text{ReLU}(-M(1 - \tilde{1}_i(h)) + \mu_i) - \text{ReLU}(-M(1 - \tilde{1}_i(h)) - \mu_i) \right). \tag{5}$$

Denoting

$$B := \bigcup_{i=1}^{k-1} [h_i - \delta_i^-, h_i + \delta_i^+]$$

note that if $h \notin B$, $\forall z$, $f(h, z) = \tilde{f}(h, z)$, as desired. If $h \in [h_i - \delta_i^-, h_i + \delta_i^+]$, $f(h, z)$ by construction will be $\gamma z + \sum_{i=1}^k w_i(h) \mu_i$ for some $w_i(h) \in [0, 1]$ s.t. $\sum_i w_i(h) = 1$.

Denoting by $\phi(h, z)$ the joint pdf of $h, z$, by the coupling definition of $W_1$, we have

$$
\begin{aligned}
W_1(f_{\#\mathcal{N}}, \mu) &\le \int_{h \in \mathbb{R}, z \in \mathbb{R}^d} \left| \tilde{f}(h, z) - f(h, z) \right|_1 d\phi(h, z) \\
&= \int_{h \in \mathbb{R}} \left| \sum_{i=1}^k \mathbb{1}(h \in (h_{i-1}, h_i]) \mu_i - \sum_{i=1}^k \left( \text{ReLU}(-M(1 - \tilde{1}_i(h)) + \mu_i) - \right. \right. \\
&\qquad \left. \left. \text{ReLU}(-M(1 - \tilde{1}_i(h)) - \mu_i)) \right) \right|_1 d\phi(h) \\
&= \int_{h \in B} \left| \sum_{i=1}^k \mathbb{1}(h \in (h_{i-1}, h_i]) \mu_i - \sum_i w_i(h) \mu_i \right|_1 d\phi(h) \\
&\le \int_{h \in B} \max_{i,j} |\mu_i - \mu_j|_1 d\phi(h) \\
&= \int_{h \in B} 2M\sqrt{d} d\phi(h) \\
&= 2M\sqrt{d} \Pr[h \in B] \\
&= 2M\sqrt{d} k\delta
\end{aligned}
$$

---

[6]The size of $g$ doesn't indeed depend on $\epsilon$. The weights in the networks will simply grow with $\epsilon$.

So if we choose $\delta = \frac{\epsilon}{2M\sqrt{dk}}$, we have the desired bound in $W_1$. (We note, making $\delta$ small only manifests in the size of the weights of the functions $\tilde{1}$, and not in the size of the network itself. This is obvious from the construction in Lemma 3 in Arora et al. (2017).)

Proceeding to subgaussianity, consider a 1-Lipschitz function $\varphi$ centered such that $\mathbb{E}[(\varphi \circ f)_{\#\mathcal{N}}] = 0$. Next, we'll show that $(\varphi \circ f)_{\#\mathcal{N}}$ is subgaussian with an appropriate constant. We can view $f_{\#\mathcal{N}}$ as the sum of two random variables: $\gamma z$ and

$$\sum_{i=1}^{k} \left( \mathrm{ReLU}(-M(1 - \tilde{1}_i(h)) + \mu_i) - \mathrm{ReLU}(-M(1 - \tilde{1}_i(h)) - \mu_i)) \right).$$

$\gamma z$ is a Gaussian with covariance $\gamma^2 I$. The other term is contained in an $l_2$ ball of radius $M$. Using the Lipschitz property and Lipschitz concentration for Gaussians (Theorem 5.2.2 of Vershynin (2018)), we see that $\Pr[|(\varphi \circ f)| \geq t] \leq \exp\left( -\frac{(t-M)^2}{2\gamma^2} \right)$. By considering separately the cases $|t| \leq 2M$ and $|t| > 2M$, we immediately see this implies that the pushforward is $O(\gamma^2 + M^2)$-subgaussian. Since $M^2 = O(\gamma^2 d)$, the claim follows. $\qquad\square$

### A.5 KL DIVERGENCE BOUNDS

In this section, we use the Bobkov-Goetze inequality to derive the KL divergence bounds from the Wasserstein bounds.

Concretely:

**Theorem 8** (Bobkov & Götze (1999))**.** *Let $p, q : \mathbb{R}^d \to \mathbb{R}^+$ be two distributions s.t. for every 1-Lipschitz $f : \mathbb{R}^d \to \mathbb{R}^+$ and $X \sim p$, $f(X)$ is $c^2$-subgaussian. Then, we have $KL(q, p) \geq \frac{1}{2c^2} W_1(p, q)^2$.*

Then, to finish the two inequalities in the statement of the main theorem, we will show that:

- For any mixture of $k$ Gaussians where the component means $\mu_i$ satisfy $\|\mu_i\| \leq M$, the condition of Theorem 8 is satisfied with $c^2 = O(\gamma^2 + M^2)$. (In fact, we show this for the pushforward through $g$, the neural network which approximates the mixture, which poses some non-trivial technical challenges. See Appendix A, Lemma 9.)
- A pushforward of the standard Gaussian through a $L$-Lipschitz generator $f$ satisfies the conditions of Theorem 8 with $c^2 = L^2$, which implies the second part of the claim. (Theorem 5.2.2 in Vershynin (2018).)

## B MISSING PROOFS FOR SECTION 5

### B.1 UPPER BOUND

First, we recall a folklore result about permutations. Let $S_n$ denote the symmetric group on $n$ elements, i.e. the set of permutations of $\{1, \ldots, n\}$ equipped with the multiplication operation of composition. Recall that the *order* of a permutation $\pi$ is the smallest positive integer $k$ such that $\pi^k$ is the identity permutation.

**Lemma 10.** *For any permutation $\pi \in S_n$, there exists $\sigma_1, \sigma_2 \in S_n$ of order at most 2 such that*

$$\pi = \sigma_1 \sigma_2.$$

*Proof.* This result is folklore. We include a proof of it for completeness[7].

First, recall that every permutation $\pi$ has a unique decomposition $\pi = c_1 \cdots c_k$ as a product of disjoint cycles. Therefore if we show the result for a single cycle, so $c_i = \sigma_{i1}\sigma_{i2}$ for every $i$, then

---

[7]This proof, given by HH Rugh, and some other ways to prove this result can be found at `https://math.stackexchange.com/questions/1871783/` `every-permutation-is-a-product-of-two-permutations-of-order-2`.

taking $\sigma_1 = \prod_{i=1}^{k} \sigma_{i1}$ and $\sigma_2 = \prod_{i=1}^{k} \sigma_{i2}$ proves the desired result since $\pi = \sigma_1 \sigma_2$ and $\sigma_1, \sigma_2$ are both of order at most 2.

It remains to prove the result for a single cycle $c$ of length $r$. The cases $r \leq 2$ are trivial. Without loss of generality, we assume $c = (1 \cdots r)$. Let $\sigma_1(1) = 2$, $\sigma_1(2) = 1$, and otherwise $\sigma_1(s) = r + 3 - s$. Let $\sigma_2(1) = 3$, $\sigma_2(2) = 2$, $\sigma_2(3) = 1$, and otherwise $\sigma_2(s) = r + 4 - s$. It's easy to check from the definition that both of these elements are order at most 2.

We now claim $c = \sigma_2 \circ \sigma_1$. To see this, we consider the following cases:

1. $\sigma_2(\sigma_1(1)) = \sigma_2(2) = 2$.

2. $\sigma_2(\sigma_1(2)) = \sigma_2(1) = 3$.

3. $\sigma_2(\sigma_1(r)) = \sigma_2(3) = 1$.

4. For all other $s$, $\sigma_2(\sigma_1(s)) = \sigma_2(r + 3 - s) = s + 1$.

In all cases we see that $c(s) = \sigma_2(\sigma_1(s))$ which proves the result. $\qquad \square$

Next, we supply the proof of Lemma 2

*Proof of Lemma 2.* It is easy to see that swapping two elements is possible in a fashion that doesn't affect other dimensions by the following 'signed swap' procedure requiring 3 matrices:

$$(x, y) \mapsto (x, y - x) \mapsto (y, y - x) \mapsto (y, -x). \tag{6}$$

Next, let $L = \{1, \ldots, d\}$ and $R = \{d + 1, \ldots, 2d\}$. There will be an equal number of elements which in a particular permutation will be permuted from $L$ to $R$ as those which will be permuted from $R$ to $L$. We can choose an arbitrary bijection between the two sets of elements and perform these 'signed swaps' in parallel as they are disjoint, using a total of 3 matrices. The result of this will be the elements partitioned into $L$ and $R$ that would need to be mapped there.

We can also (up to sign) transpose elements within a given set $L$ or $R$ via the following computation using our previous 'signed swaps' that requires one 'storage component' in the other set:

$$([x, y], z) \mapsto ([z, y], -x) \mapsto ([z, x], y) \mapsto ([y, x], -z).$$

So, up to sign, we can in 9 matrices compute any transposition in L or R separately. In fact, since any permutation can be represented as the product of two order-2 permutations (Lemma 10) and any order-2 permutation is a disjoint union of transpositions, we can implement an order-2 permutation up to sign using 9 matrices and an arbitrary permutation up to sign using 18 matrices.

In total, we used 3 matrices to move elements to the correct side and 18 matrices to move them to their correct position, for a total of 21 matrices. $\qquad \square$

**Lemma 11.** *Suppose $A \in \mathbb{R}^{n \times n}$ is a matrix with $n$ distinct real eigenvalues. Then there exists an invertible matrix $S \in \mathbb{R}^{n \times n}$ such that $A = SDS^{-1}$ where $D$ is a diagonal matrix containing the eigenvalues of $A$.*

*Proof.* Observe that for every eigenvalue $\lambda_i$ of $A$, the matrix $(A - \lambda_i I)$ has rank $n - 1$ by definition, hence there exists a corresponding real eigenvector $v_i$ by taking a nonzero solution of the real linear system $(A - \lambda I)v = 0$. Taking $S$ to be the linear operator which maps $e_i$ to standard basis vector $v_i$, and $D = \text{diag}(\lambda_1, \ldots, \lambda_n)$ proves the result. $\qquad \square$

Next, we give the proof of Lemma 3

*Proof of Lemma 3.* Let

$$D = (M - I)E^{-1},$$
$$H = (M^{-1} - I)E^{-1},$$
$$E = -AM,$$

where $A$ is an invertible matrix that will be specified later. We can multiply out with these values giving

$$\begin{bmatrix} I & 0 \\ A & I \end{bmatrix} \begin{bmatrix} I & D \\ 0 & I \end{bmatrix} \begin{bmatrix} I & 0 \\ E & I \end{bmatrix} \begin{bmatrix} I & H \\ 0 & I \end{bmatrix}$$

$$= \begin{bmatrix} I & 0 \\ A & I \end{bmatrix} \begin{bmatrix} I & (I-M)M^{-1}A^{-1} \\ 0 & I \end{bmatrix} \begin{bmatrix} I & 0 \\ -AM & I \end{bmatrix} \begin{bmatrix} I & (I-M^{-1})M^{-1}A^{-1} \\ 0 & I \end{bmatrix}$$

$$= \begin{bmatrix} I & (M^{-1}-I)A^{-1} \\ A & AM^{-1}A^{-1} \end{bmatrix} \begin{bmatrix} I & 0 \\ -AM & I \end{bmatrix} \begin{bmatrix} I & (I-M^{-1})M^{-1}A^{-1} \\ 0 & I \end{bmatrix}$$

$$= \begin{bmatrix} M & (M^{-1}-I)A^{-1} \\ 0 & AM^{-1}A^{-1} \end{bmatrix} \begin{bmatrix} I & (I-M^{-1})M^{-1}A^{-1} \\ 0 & I \end{bmatrix}$$

$$= \begin{bmatrix} M & 0 \\ 0 & AM^{-1}A^{-1} \end{bmatrix}$$

Here what remains is to guarantee $AM^{-1}A^{-1} = S$. Since $S$ and $M^{-1}$ have the same eigenvalues, by Lemma 11 there exist real matrices $U, V$ such that $S = UXU^{-1}$ and $M^{-1} = VXV^{-1}$ for the same diagonal matrix $X$, hence $S = UV^{-1}M^{-1}VU^{-1}$. Therefore taking $A = UV^{-1}$ gives the result. $\qquad\square$

Now that we have the Lemmas, we prove the upper bound.

*Proof of Theorem 5.* Recall that our goal is to show that $GL_+(2d, \mathbb{R}) \subset \mathcal{A}^K$ for an absolute constant $K > 0$. To show this, we consider an arbitrary matrix $T \in GL_+(2d, \mathbb{R})$, i.e. an arbitrary matrix $T : 2d \times 2d$ with positive determinant, and show how to build it as a product of a bounded number of elements from $\mathcal{A}$. As $T$ is a square matrix, it admits an LUP decomposition Horn & Johnson (2012): i.e. a decomposition into the product of a lower triangular matrix $L$, an upper triangular matrix $U$, and a permutation matrix $P$. This proof proceeds essentially by showing how to construct the $L, U$, and $P$ components in a constant number of our desired matrices.

By Lemma 2, we can produce a matrix $\tilde{P}$ with $\det \tilde{P} > 0$ which agrees with $P$ up to the sign of its entries using $O(1)$ linear affine coupling layers. Then $T\tilde{P}^{-1}$ is a matrix which admits an $LU$ decomposition: for example, given that we know $TP^{-1}$ has an $LU$ decomposition, we can modify flip the sign of some entries of $U$ to get an LU decomposition of $T\tilde{P}^{-1}$. Furthermore, since $\det(T\tilde{P}^{-1}) > 0$, we can choose an $LU$ decomposition $T\tilde{P}^{-1} = LU$ such that $\det(L), \det(U) > 0$ (for any decomposition which does not satisfy this, the two matrices $L$ and $U$ must both have negative determinant as $0 < \det(T\tilde{P}^{-1}) = \det(L)\det(U)$. In this case, we can flip the sign of column $i$ in $L$ and row $i$ in $U$ to make the two matrices positive determinant).

It remains to show how to construct a lower/upper triangular matrix with positive determinant out of our matrices. We show how to build such a lower triangular matrix $L$ as building $U$ is symmetrical.

At this point we have a matrix $\begin{bmatrix} A & 0 \\ B & C \end{bmatrix}$, where $A$ and $C$ are lower triangular. We can use column elimination to eliminate the bottom-left block:

$$\begin{bmatrix} A & 0 \\ B & C \end{bmatrix} \begin{bmatrix} I & 0 \\ -C^{-1}B & I \end{bmatrix} = \begin{bmatrix} A & 0 \\ 0 & C \end{bmatrix},$$

where $A$ and $C$ are lower-triangular.

Recall from equation 6 that we can perform the signed swap operation in $\mathbb{R}^2$ of taking $(x, y) \mapsto (y, -x)$ for $x$ using 3 affine coupling blocks. Therefore using 6 affine coupling blocks we can perform a sign flip map $(x, y) \mapsto (-x, -y)$. Note that because $\det(L) > 0$, the number of negative entries in the first $d$ diagonal entries has the same parity as the number of negative entries in the second $d$ diagonal entries. Therefore, using these sign flips in parallel, we can ensure using 6 affine coupling layers that that the first $d$ and last $d$ diagonal entries of $L$ have the same number of negative elements. Now that the number of negative entries match, we can apply two diagonal rescalings to ensure that:

1. The first $d$ diagonal entries of the matrix are distinct.

2. The last $d$ diagonal entries contain the multiplicative inverses of the first $d$ entries up to reordering. Here we use that the number of negative elements in the first $d$ and last $d$ elements are the same, which we ensured earlier.

At this point, we can apply Lemma 3 to construct this matrix from four of our desired matrices. Since this shows we can build $L$ and $U$, this shows we can build any matrix with positive determinant.

Now, let's count the matrices we needed to accomplish this. In order to construct $\tilde{P}$, we needed 21 matrices. To construct $L$, we needed 1 for column elimination, 6 for the sign flip, 2 for the rescaling of diagonal elements, and 4 for Lemma 3 giving a total of 13. So, we need $21 + 13 + 13 = 47$ total matrices to construct the whole $LUP$ decomposition. □

## B.2 LOWER BOUND

Finally, we proceed to give the proof of Lemma 4.

*Proof of Lemma 4.* We explicitly solve the block matrix equations. Multiplying out the LHS gives

$$\begin{bmatrix} C & D \\ AC & AD+B \end{bmatrix} \begin{bmatrix} G & H \\ EG & EH+F \end{bmatrix} = \begin{bmatrix} CG+DEG & CH+DEH+DF \\ ACG+ADEG+BEG & ACH+ADEH+ADF+BEH+BF \end{bmatrix}.$$

Say

$$T = \begin{bmatrix} X & Y \\ Z & W \end{bmatrix}.$$

Starting with the top-left block gives that

$$X = (C+DE)G$$

$$D = (XG^{-1}-C)E^{-1} \tag{7}$$

Next, the top-right block gives that

$$Y = (C+DE)H + DF = XG^{-1}H + DF$$

$$H = GX^{-1}(Y-DF). \tag{8}$$

Equivalently,

$$D = (Y - XG^{-1}H)F^{-1} \tag{9}$$

Combining equation 8 and equation 7 gives

$$H = GX^{-1}(Y - (XG^{-1}-C)E^{-1}F)$$

$$H = GX^{-1}Y - (I - GX^{-1}C)E^{-1}F \tag{10}$$

The bottom-left and equation 7 gives

$$Z = ACG + ADEG + BEG$$

$$ZG^{-1} = AC + (AD+B)E$$

$$E = (AD+B)^{-1}(ZG^{-1}-AC) \tag{11}$$

$$E = (A(XG^{-1}-C)E^{-1}+B)^{-1}(ZG^{-1}-AC)$$

$$E^{-1} = (ZG^{-1}-AC)^{-1}(A(XG^{-1}-C)E^{-1}+B)$$

$$(ZG^{-1}-AC) = (A(XG^{-1}-C)E^{-1}+B)E = A(XG^{-1}-C)+BE$$

$$E = B^{-1}((ZG^{-1}-AC)-A(XG^{-1}-C))$$

$$E = B^{-1}(ZG^{-1}-AXG^{-1}) \tag{12}$$

Taking the bottom-right block and substituting equation 11 gives

$$W = ACH + (AD+B)(EH+F) = ACH + (ZG^{-1}-AC)H + (AD+B)F$$

$$W = ZG^{-1}H + ADF + BF. \tag{13}$$

Substituting equation 7 into equation 13 gives

$$W = ZG^{-1}H + A(Y - XG^{-1}H) + BF = (Z - AX)G^{-1}H + AY + BF.$$

Substituting equation 10 gives

$$\begin{aligned} W &= (Z - AX)G^{-1}(GX^{-1}Y - (I - GX^{-1}C)E^{-1}F) + AY + BF \\ &= (Z - AX)(X^{-1}Y - (G^{-1} - X^{-1}C)E^{-1}F) + AY + BF. \end{aligned}$$

Substituting equation 12 gives

$$W = (Z - AX)(X^{-1}Y - (G^{-1} - X^{-1}C)(ZG^{-1} - AXG^{-1})^{-1}BF) + AY + BF$$

$$\begin{aligned} W - ZX^{-1}Y - BF &= (Z - AX)(X^{-1}C - G^{-1})((Z - AX)G^{-1})^{-1}BF \\ &= (Z - AX)(X^{-1}C - G^{-1})G(Z - AX)^{-1}BF \\ &= (Z - AX)X^{-1}C(Z - AX)^{-1} - BF \end{aligned}$$

$$W - ZX^{-1}Y = (Z - AX)X^{-1}C(Z - AX)^{-1} \tag{14}$$

Here we notice that $W - ZX^{-1}Y$ is similar to $X^{-1}C$, where we get to choose values along the diagonal of $C$. In particular, this means that $W - ZX^{-1}Y$ and $X^{-1}C$ must have the same eigenvalues. $\qquad\square$

*Proof of Theorem 6.* First, note that element in $\mathcal{A}^4$ can be written in either the form $L_1 R_1 L_2 R_2$ or $R_1 L_1 R_2 L_2$ for $L_1, L_2 \in \mathcal{A}_\mathcal{L}$ and $R_1, R_2 \in \mathcal{A}_\mathcal{R}$. We construct an explicit matrix which cannot be written in either form.

Consider an invertible matrix of the form

$$T = \begin{bmatrix} X & 0 \\ 0 & W \end{bmatrix}$$

and observe that the Schur complement $T/X$ is simply $W$. Therefore Lemma 4 says that this matrix can only be in $\mathcal{A}_L \mathcal{A}_R \mathcal{A}_L \mathcal{A}_R$ if $W$ is similar to $X^{-1}C$ for some diagonal matrix $C$. Now consider the case where $W$ is a permutation matrix encoding the permutation $(1\ 2\ \cdots\ d)$ and $X$ is a diagonal matrix with nonzero entries. Then $X^{-1}C$ is a diagonal matrix as well, hence has real eigenvalues, while the eigenvalues of $W$ are the $d$-roots of unity. (The latter claim follows because for any $\zeta$ with $\zeta^d = 1$, the vector $(1, \zeta, \cdots, \zeta^{d-1})$ is an eigenvector of $W$ with eigenvalue $\zeta$). Since similar matrices must have the same eigenvalues, it is impossible that $X^{-1}C$ and $W$ are similar.

The remaining possibility we must consider is that this matrix is in $\mathcal{A}_\mathcal{R} \mathcal{A}_\mathcal{L} \mathcal{A}_\mathcal{R} \mathcal{A}_\mathcal{L}$. In this case by applying the symmetrical version of Lemma 4 (which follows by swapping the first $n$ and last $n$ coordinates), we see that $W^{-1}C$ and $X$ must be similar. Since $\text{Tr}(W^{-1}C) = 0$ and $\text{Tr}(X) > 0$, this is impossible. $\qquad\square$

## C  EXPERIMENTAL VERIFICATION

### C.1  PARTITIONED LINEAR NETWORKS

In this section, we will provide empirical support for Theorems 2 and 3. More precisely, empirically, the number of required linear affine coupling layers at least for random matrices seems closer to the lower bound – so it's even better than the upper bound we provide.

**Setup**  We consider the following synthetic setup. We train $n$ layers of affine coupling layers, namely networks of the form

$$f_n(z) = \prod_{i=1}^{n} E_i \begin{bmatrix} C_i & D_i \\ 0 & I \end{bmatrix} \begin{bmatrix} I & 0 \\ A_i & B_i \end{bmatrix}$$

with $E_i, B_i, C_i$ diagonal. Notice the latter two follow the statement of Theorem 2 and the alternating order of upper vs lower triangular matrices can be assumed without loss of generality, as a product of upper/lower triangular matrices results in an upper/lower triangular matrix. The matrices $E_i$ turn out

to be necessary for training – they enable "renormalizing" the units in the network (in fact, Glow uses these and calls them actnorm layers; in older models like RealNVP, batchnorm layers are used instead).

The training data is of the form $Az$, where $z \sim \mathcal{N}(0, I)$ for a fixed $d \times d$ square matrix $A$ with either random standard Gaussian entries in Figures 4 to 8 or random standard Gaussian entries that are diagonal-constant (that latter giving a natural random ensemble of Toeplitz matrices) in Figures 9 to 13. This ensures that there is a "ground" truth linear model that fits the data well. [8] We then train the affine coupling network by minimizing the loss $\mathbb{E}_{z \sim \mathcal{N}(0,I)} \left[ (f_n(z) - Az)^2 \right]$ and trained on a variety of values for $n$ and $d$ in order to investigate how the depth of linear networks affects the ability to fit linear functions of varying dimension.

Note, we are not training via maximum likelihood, but rather we are minimizing a "supervised" loss, wherein the network $f_n$ "knows" which point $x$ a latent $z$ is mapped to. This is intentional and is meant to separate the representational vs training aspect of different architectures. Namely, this objective is easier to train, and our results address the representational aspects of different architectures of flow networks – so we wish our experiments to be confounded as little as possible by aspects of training dynamics.

We chose $n = 1, 2, 4, 8, 16$ layers and $d = 4, 8, 16, 32, 64$ dimensions (here a layer is one matrix and not a flipped pair as in our theoretical results). We present the standard L2 training loss and the squared Frobenius error of the recovered matrix $\hat{A}$ obtained by multiplying out the linear layers $||\hat{A} - A||_F^2$, both normalized by $1/d^2$ so that they are independent of dimensionality. We shade the standard error regions of these losses across the seeds tried. All these plots are log-scale, so the noise seen lower in the charts is very small.

We initialize the $E, C, B$ matrices with 1s on the diagonal and $A, D$ with random Gaussian elements with $\sigma = 10^{-5}$ and train with Adam with learning rate $10^{-4}$. We train on 5 random seeds which affect the matrix $A$ generated and the datapoints $z$ sampled.

Finally, we also train similar RealNVP models on the same datasets, using a regression objective as done with the PLNs but $s$ and $t$ networks with two hidden layers with 128 units and the same numbers of couplings as with the PNN experiments.

**Results** The results demonstrate that the 1- and 2- layer networks fail to fit even coarsely any of the linear functions we tried. Furthermore, the 4-layer networks consistently under-perform compared to the 8- and 16-layer networks. The 8- and 16-layer networks seem to perform comparably, though we note the larger mean error for d=64, which suggests that the performance can potentially be further improved (either by adding more layers, or improving the training by better choice of hyperparameters; even on this synthetic setup, we found training of very deep networks to be non-trivial).

These experimental results suggest that at least for random linear transformations $T$, the number of required linear layers is closer to the lower bound. Moreover, the error for the Toeplitz ensemble is slightly larger, indicating this distribution is slightly harder. Closing this gap (both in a worst-case and distributional sense) is an interesting question for further work.

In our experiments with the RealNVP architecture, we observe more difficulty in fitting these linear maps, as they seem to need more training data to reach similar levels or error. We hypothesize this is due to the larger model class that comes with allowing nonlinear functions in the couplings.

### C.2 Additional Padding Results on Synthetic Datasets

We provide further results on the performance of Real NVP models on datasets with different kinds of padding (no padding, zero-padding and Gaussian padding) on standard synthetic datasets–Swissroll, 2 Moons and Checkerboard.

The results are consistent with the performance on the mixture of 4 Gaussians: in Figures 24, 25, and 26, we see that the zero padding greatly degrades the conditioning and somewhat degrades the visual quality of the learned distribution. On the other hand, Gaussian padding consistently performs best, both in terms of conditioning of the Jacobian, and in terms of the quality of the recovered distribution.

---

[8]As a side remark, this ground truth is only specified up to orthogonal matrices $U$, as $AUz$ is identically distributed to $Az$, due to the rotational invariance of the standard Gaussian.

# D    UNIVERSAL APPROXIMATION WITH ILL-CONDITIONED AFFINE COUPLING NETWORKS

## D.1    SIMPLER UNIVERSALITY UNDER ZERO-PADDING.

First (as a warmup), we give a much simpler proof than Huang et al. (2020) that affine coupling networks are universal approximators in Wasserstein under zero-padding, which moreover shows that only a small number of affine coupling layers are required. For $Q$ a probability measure over $\mathbb{R}^n$ satisfying weak regularity conditions (see Theorem 9 below), by Brenier's Theorem Villani (2003) there a $W_2$ optimal transport map

$$\varphi : \mathbb{R}^n \to \mathbb{R}^n$$

such that if $X \sim N(0, I_{n \times n})$, then the pushforward $\varphi_\#(X)$ is distributed according to $Q$, and a corresponding transport map in the opposite direction which we denote $\varphi^{-1}$. If we allow for arbitrary functions $t$ in the affine coupling network, then we can implement the zero-padded transport map $(X, 0) \mapsto (\varphi(X), 0)$ as follows:

$$(X, 0) \mapsto (X, \varphi(X)) \mapsto (\varphi(X), \varphi(X)) \mapsto (\varphi(X), 0). \tag{15}$$

Explicitly, in the first layer the translation map is $t_1(x) = \varphi(x)$, in the second layer the translation map is $t_2(x) = x - \varphi^{-1}(x)$, and in the third layer the translation map is $t_3(x) = -x$. Note that no scaling maps are required: with zero-padding the basic NICE architecture can be universal, unlike in the unpadded case where NICE can only hope to implement volume preserving maps. This is because every map from zero-padded data to zero-padded data is volume preserving. Finally, if we are required to implement the translation maps using neural networks, we can use standard approximation-theoretic results for neural networks, combined with standard results from optimal transport, to show universality of affine coupling networks in Wasserstein. First, we recall the formal statement of Brenier's Theorem:

**Theorem 9** (Brenier's Theorem, Theorem 2.12 of Villani (2003))**.** *Suppose that $P$ and $Q$ are probability measures on $\mathbb{R}^n$ with densities with respect to the Lebesgue measure. Then $Q = (\nabla \psi)_\# P$ for $\psi$ a convex function, and moreover $\nabla \psi$ is the unique $W_2$-optimal transport map from $P$ to $Q$.*

It turns out that the transportation map $\varphi := \nabla \psi$ is not always a continuous function, however there are simple sufficient conditions for the distribution $Q$ under which the map is continuous (see e.g. Caffarelli (1992)). From these results (or by directly smoothing the transport map), we know any distribution with bounded support can be approached in Wasserstein distance by smooth pushforwards of Gaussians. So for simplicity, we state the following Theorem for distributions which are the pushforward of smooth maps.

**Theorem 10** (Universal approximation with zero-padding)**.** *Suppose that $P$ is the standard Gaussian measure in $\mathbb{R}^n$ and $Q = \varphi_\# P$ is the pushforward of the Gaussian measure through $\varphi$ and $\varphi$ is a smooth map. Then for any $\epsilon > 0$ there exists a depth 3 affine coupling network $g$ with no scaling and feedforward ReLU net translation maps such that $W_2(g_\#(P \times \delta_{0^n}), Q \times \delta_{0^n}) \le \epsilon$.*

*Proof.* For any $M > 0$, let $f_M(x) = \min(M, \max(-M, x))$ be the 1-dimensional truncation map to $[-M, M]$ and for a vector $x \in \mathbb{R}^n$ let $f_M(x) \in [-M, M]^n$ be the result of applying $f_M$ coordinate-wise. Note that $f_M$ can be implemented as a ReLU network with two hidden units per input dimension. Also, any continuous function on $[-M, M]^n$ can be approximated arbitrarily well in $L^\infty$ by a sufficiently large ReLU neural network with one hidden layer Leshno et al. (1993). Finally, note that if $\|f - g\|_{L^\infty} \le \epsilon$ then for any distribution $P$ we have $W_2(f_\# P, g_\# P) \le \epsilon$ by considering the natural coupling that feeds the same input into $f$ and $g$.

Now we show how to approximate the construction of equation 15 using these tools. For any $\epsilon > 0$, if we choose $M$ sufficiently large and then take $\tilde{\varphi}$ and $\widetilde{\varphi^{-1}}$ to be sufficiently good approximations of $\varphi$ and $\varphi^{-1}$ on $[-M, M]^n$, we can construct an affine coupling network with ReLU feedforward network translation maps $\tilde{t}_1(x) = f_M(\tilde{\varphi}(f_M(x)))$, $\tilde{t}_2(x) = x - \widetilde{\varphi^{-1}}(x)$, and $\tilde{t}_3(x) = -x$, such that the output has $W_2$ distance at most $\epsilon$ from $Q$. □

**Universality without padding.**    We now show that universality in Wasserstein can be proved even if we don't have zero-padding, using a lattice-based encoding and decoding scheme. Let $\epsilon > 0$

be a small constant, to be taking sufficiently small. Let $\epsilon' \in (0, \epsilon)$ be a further constant, taken sufficiently small with respect to $\epsilon$ and similar for $\epsilon''$ wrt $\epsilon'$. Suppose the input dimension is $2n$, and let $X = (X_1, X_2)$ with independent $X_1 \sim N(0, I_{n \times n})$ and $X_2 \sim N(0, I_{n \times n})$ be the input the the affine coupling network. Let $f(x)$ be the map which rounds $x \in \mathbb{R}^n$ to the closest grid point in $\epsilon \mathbb{Z}^n$ and define $g(x) = x - f(x)$. Note that for a point of the form $z = f(x) + \epsilon' y$ for $y$ which is not too large, we have that $f(z) = f(x)$ and $g(z) = y$. Let $\varphi_1, \varphi_2$ be the desired transportation maps guaranteed by Brenier's theorem, so that the distribution of $\varphi_1(X)$ is the target distribution $Q$ and $\varphi_2(X)$ is a standard Gaussian independent of $\varphi_1(X)$. (In other words, $\varphi_1, \varphi_2$ correspond to the first half and second half of the output coordinates of the transport map from the $2n$ dimensional standard Gaussian to the desired padded distribution.) Now we consider the following sequence of maps:

$$(X_1, X_2) \mapsto (X_1, \epsilon' X_2 + f(X_1)) \tag{16}$$
$$\mapsto (f(\varphi_1(f(X_1), X_2)) + \epsilon' \varphi_2(f(X_1), X_2) + O(\epsilon''), \epsilon' X_2 + f(X_1)) \tag{17}$$
$$\mapsto (f(\varphi_1(f(X_1), X_2)) + \epsilon' \varphi_2(f(X_1), X_2) + O(\epsilon''), \varphi_2(f(X_1), X_2) + O(\epsilon''/\epsilon')). \tag{18}$$

More explicitly, in the first step we take $s_1(x) = \log(\epsilon') \vec{1}$ and $t_1(x) = f(x)$. In the second step, we take $s_2(x) = \log(\epsilon'') \vec{1}$ and $t_2$ is defined by $t_2(x) = f(\varphi_1(f(x), g(x))) + \epsilon' \varphi_2(f(x), g(x))$. In the third step, we take $s_3(x) = \log(\epsilon'') \vec{1}$ and define $t_3(x) = \frac{g(x)}{\epsilon'}$.

Again, taking sufficiently good approximations to all of the maps allows to approximate this map with neural networks, which we formalize below.

*Proof of Theorem 4.* Turning equation 16, equation 17, and equation 18 into a universal approximation theorem for ReLU-net based feedforward networks just requires to modify the proof of Theorem 10 for this scenario.

Fix $\delta > 0$, the above argument shows we can choose $\epsilon, \epsilon', \epsilon'' > 0$ sufficiently small so that if $h$ is map defined by composing equation 16, equation 17, and equation 18, then $W_2(h_\# P, Q) \leq \epsilon/4$. The layers defining $h$ may not be continuous, since $f$ is only continuous almost everywhere. Using that continuous functions are dense in $L^2$, we can find a function $f_\epsilon$ which is continuous and such that if we define $h_\epsilon$ by replacing each application of $f$ by $f_\epsilon$, then $W_2(h_\epsilon \# P, Q) \leq \epsilon/2$.

Finally, since $f_\epsilon$ is an affine coupling network with continuous $s$ and $t$ functions, we can use the same truncation-and-approximation argument from Theorem 10 to approximate it by an affine coupling network $g$ with ReLU feedforward $s$ and $t$ functions such that $W_2(g \# P, Q) \leq \epsilon$, which proves the result. $\square$

## E   APPROXIMATING ENTRYWISE NONLINEARITY WITH AFFINE COUPLINGS

To show how surprisingly hard it may be to represent even simple functions using affine couplings, we show an example of a very simple function—an entrywise application of hyperbolic tangent, s.t. an arbitrary depth/width sequence of affine coupling blocks with $\tanh$ nonlinearities cannot exactly represent it. Thus, even for simple functions, the affine-coupling structure imposes nontrivial restrictions. Note that in contrast to Theorem 4, we are considering exact representation here.

Precisely, we show:

**Theorem 11.** *Let $d \geq 2$ and denote $g : \mathbb{R}^d \to \mathbb{R}^d$, $g(z) := (\tanh z_1, \ldots, \tanh z_d)$. Then, for any $W, D, N \in \mathbb{N}$ and norm $\| \cdot \|$, there exists an $\varepsilon(W, D, N) > 0$, s.t. for any network $f$ consisting of a sequence of at most $N$ affine coupling layers of the form:*

$$(y_S, y_{\bar{S}}) \to (y_S, y_{\bar{S}} \odot a(y_S) + b(y_S))$$

*for in each layer an arbitrary set $S \subsetneq [d]$ and $a, b$ arbitrary feed-forward $\tanh$ neural networks of width at most $W$, depth at most $D$, and weight norm into each unit of at most $R$, it holds that*

$$\mathbb{E}_{x \in [-1,1]^d} \| f(x) - g(x) \| > \varepsilon(W, D, N, R).$$

The proof of the theorem is fairly unusual, as it uses some tools from complex analysis in several variables (see Grauert & Fritzsche (2012) for a reference) — though it's so short that we include it here. The result also generalizes to other neural networks with analytic activations.

*Proof of Theorem 11.* By compactness of the class of models bounded by $W, D, N, R$, it suffices to prove that there is no way to exactly represent the function.

Suppose for contradiction that $f = g$ on the entirety of $[-1, 1]^d$. Let $z_1, \ldots, z_d$ denote the $d$ inputs to the function: we now consider the behavior of $f$ and $g$ when we extend their definition to $\mathbb{C}^d$. From the definition, $g$ extends to a holomorphic function (of several variables) on all of $\mathbb{C}^d \setminus \{z : \exists j, z_j = i\pi(k + 1/2) : k \in \mathbb{Z}\}$, i.e. everywhere where $\tanh$ doesn't have a pole. Similarly, there exists an dense open subset $D \subset \mathbb{C}^d$ on which the affine coupling network $f$ is holomorphic, because it is formed by the addition, multiplication, and composition of holomorphic functions.

We next prove that $f = g$ on their complex extensions by the Identity Theorem (Theorem 4.1 of Grauert & Fritzsche (2012)). We must first show that $f = g$ on an open subset of $\mathbb{C}^d$. To prove this, observe that $f$ is analytic at zero and its power series expansion is uniquely defined in terms of the values of $f$ on $\mathbb{R}^d$ (for example, we can compute the coefficients by taking partial derivatives). It follows that the power series expansions of $f$ and $g$ are both equal at zero and convergent in an open neighborhood of $0$ in $\mathbb{C}^d$, so we can indeed apply the Identity Theorem; this shows that $f = g$ wherever they are both defined.

From the definition $\tanh(z) = \frac{e^{2z}-1}{e^{2z}+1}$ we can see that $g$ is periodic in the sense that $g(z + \pi i k) = g(z)$ for any $k \in \mathbb{Z}^d$. However, by construction the affine coupling network $f$ is invertible whenever, at every layer, the output of the function $a$ is not equal to zero. By the identity theorem, the set of inputs where each $a$ vanishes is nowhere dense — otherwise, by continuity $a$ vanishes on the open neighborhood of some point, so $a = 0$ by the Identity Theorem which contradicts the assumption. Therefore the union of inputs where $a$ at any layer vanishes is also nowhere dense. Consider the behavior of $f$ on an open neighborhood of $0$ and of $i\pi$: we have shown that $f$ is invertible except on a nowhere dense set, and also that $g = f$ wherever $f$ is defined, but $g(z) = g(z + i\pi)$ so it's impossible for $f$ to be invertible on these neighborhoods except on a nowhere dense subset. By contradiction, $f \neq g$ on $[-1, 1]^d$. $\qquad\square$

Finally, to give empirical evidence that the above is not merely a theoretical artifact, we regress an affine coupling architecture to fit entrywise $\tanh$.

Specifically, we sample 10-dimensional vectors from a standard Gaussian distribution and train networks as in the padding section on a squared error objective such that each input is regressed on its elementwise $\tanh$. We train an affine coupling network with 5 pairs of alternating couplings with $g$ and $h$ networks consisting of 2 hidden layers with 128 units each. For comparison, we also regress a simple MLP with 2 hidden layers with 128 units in each layer, exactly one of the $g$ or $h$ subnetworks from the coupling architecture, which contains 20 such subnetworks. For another comparison, we also try this on the elementwise ReLU function, using affine couplings with $\tanh$ activations and the same small MLP.

As we see in Figure 3, the affine couplings fit the function substantially worse than a much smaller MLP – corroborating our theoretical result.

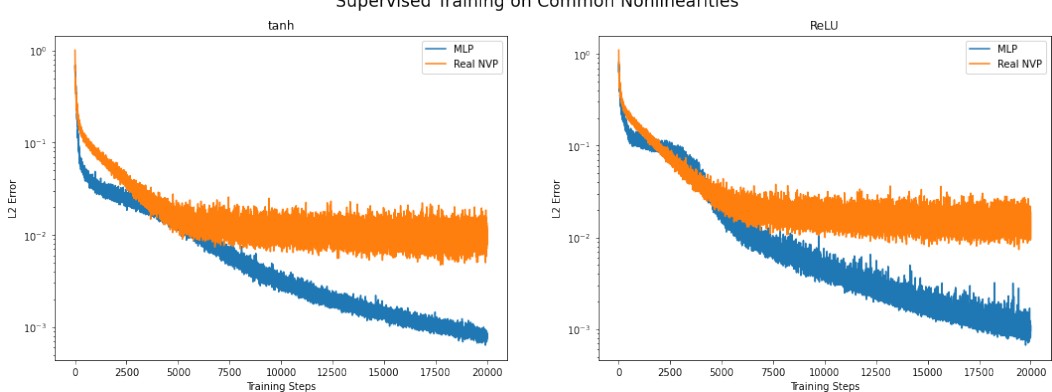

Figure 3: The smaller MLPs are much better able to fit simple elementwise nonlinearities than the affine couplings.

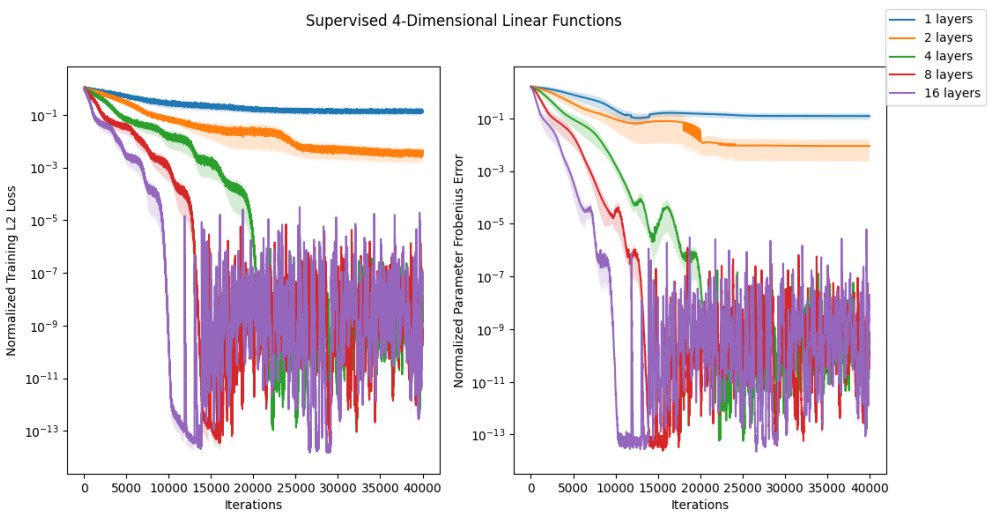

Figure 4: Learning Partitioned Linear Networks on 4-D linear functions.

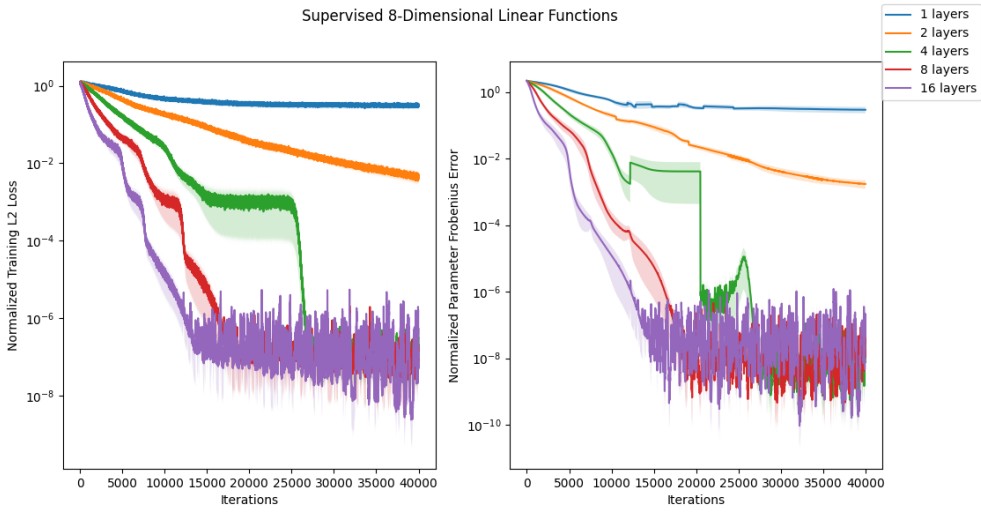

Figure 5: Learning Partitioned Linear Networks on 8-D linear functions.

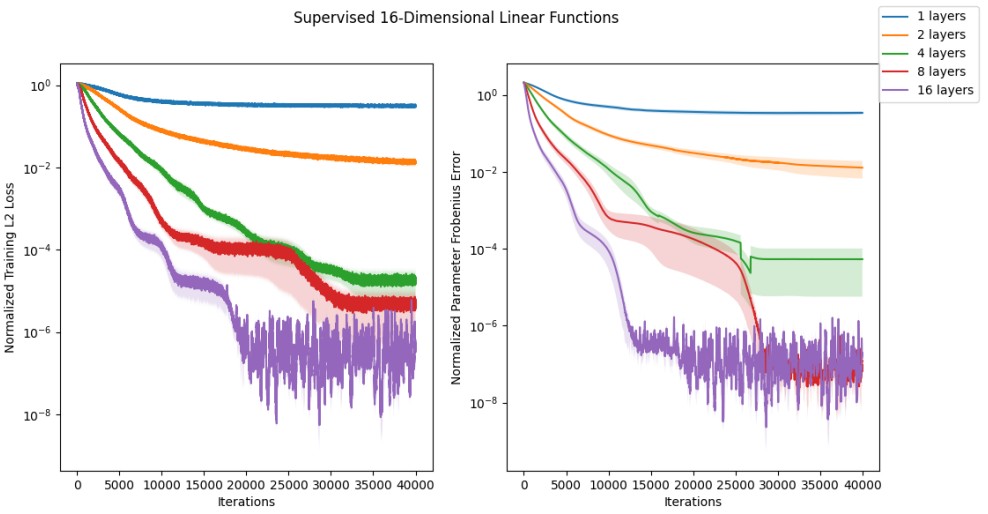

Figure 6: Learning Partitioned Linear Networks on 16-D linear functions.

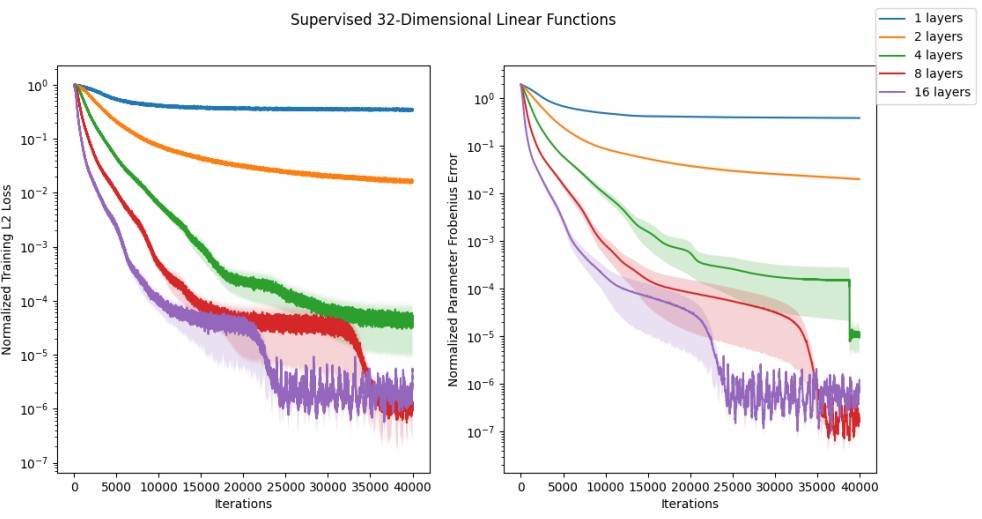

Figure 7: Learning Partitioned Linear Networks on 32-D linear functions.

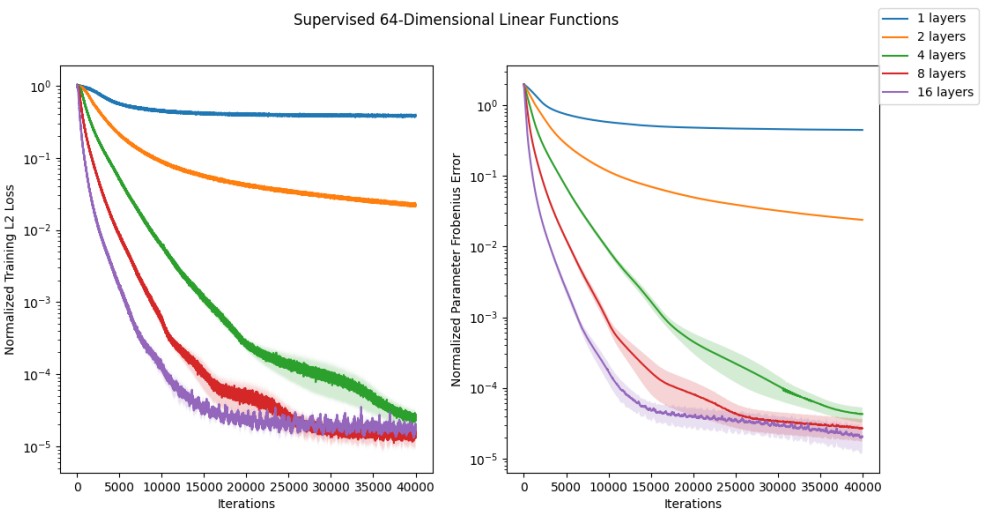

Figure 8: Learning Partitioned Linear Networks on 64-D linear functions.

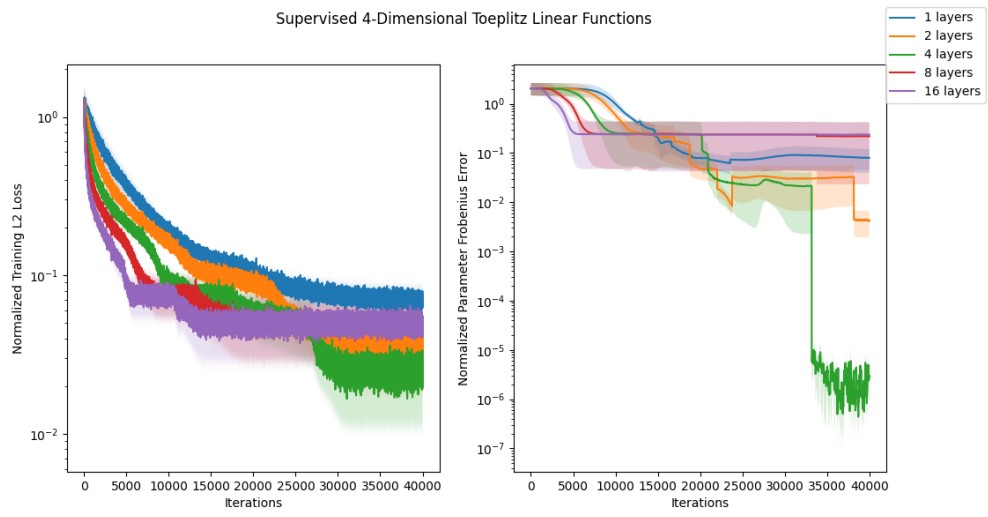

Figure 9: Learning Partitioned Linear Networks on 4-D Toeplitz functions.

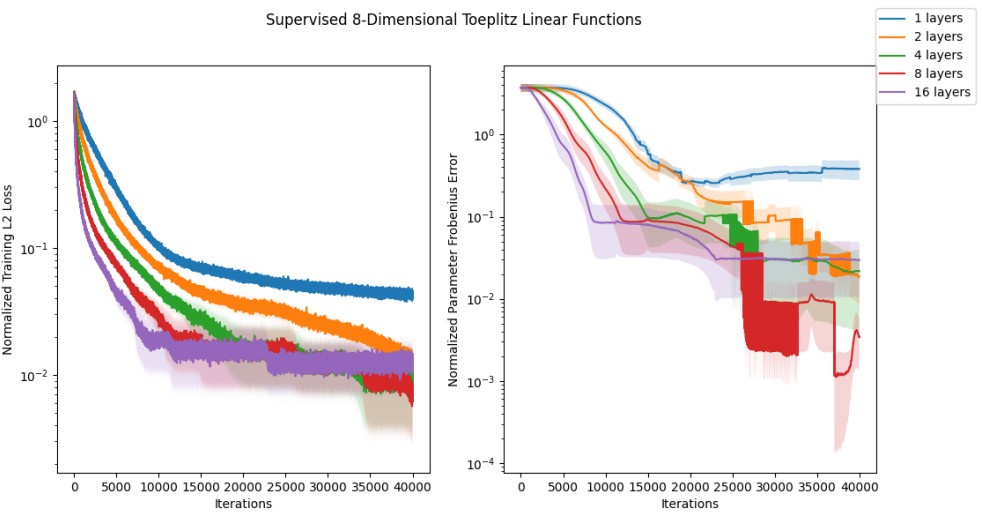

Figure 10: Learning Partitioned Linear Networks on 8-D Toeplitz functions.

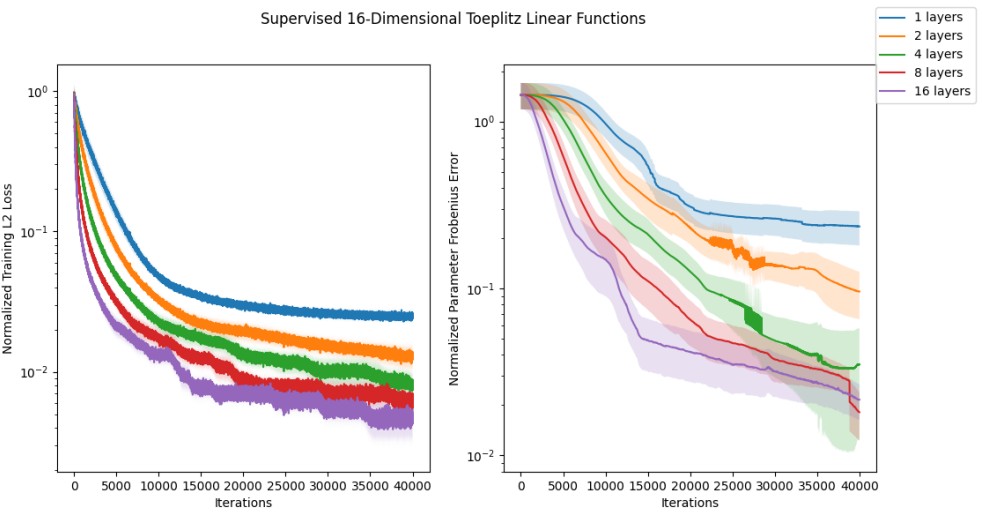

Figure 11: Learning Partitioned Linear Networks on 16-D Toeplitz functions.

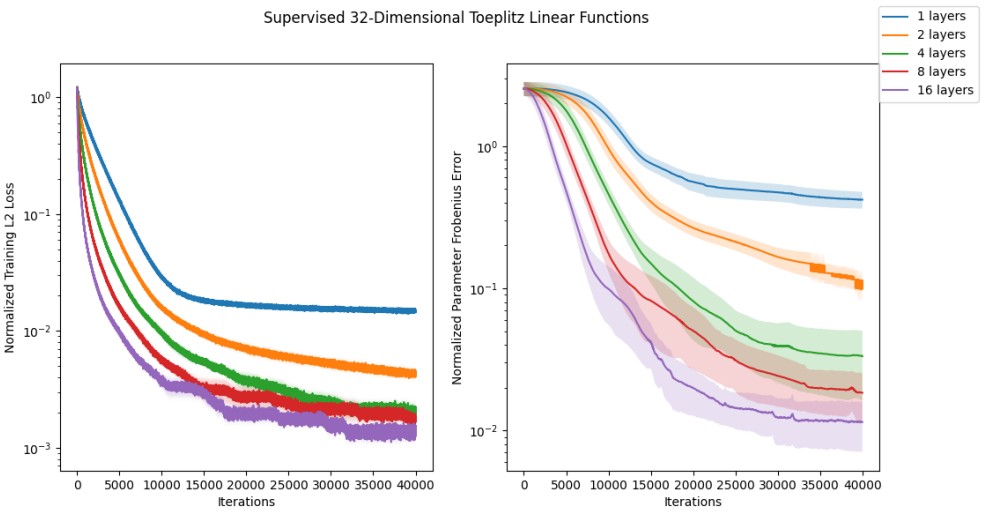

Figure 12: Learning Partitioned Linear Networks on 32-D Toeplitz functions.

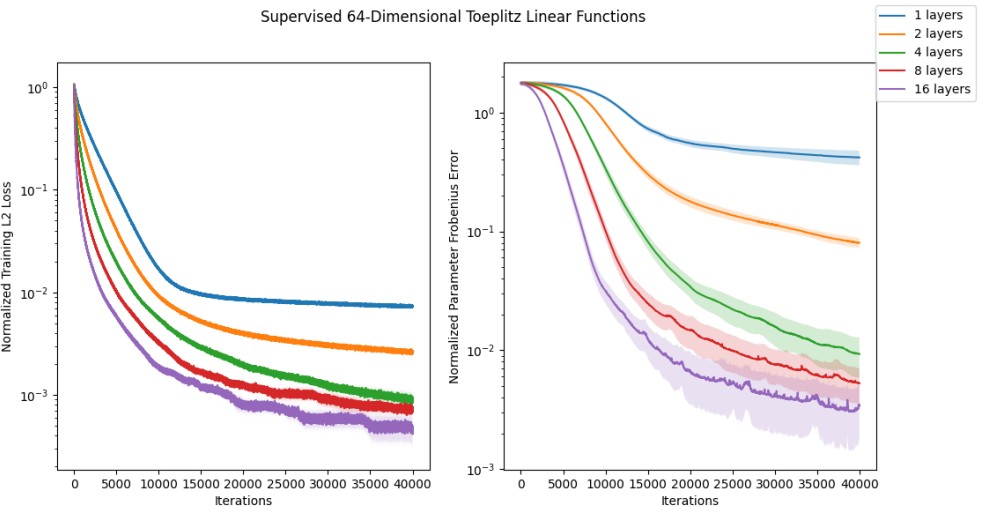

Figure 13: Learning Partitioned Linear Networks on 64-D Toeplitz functions.

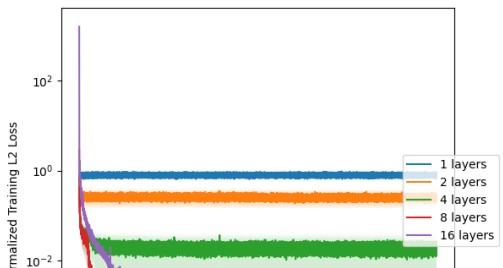

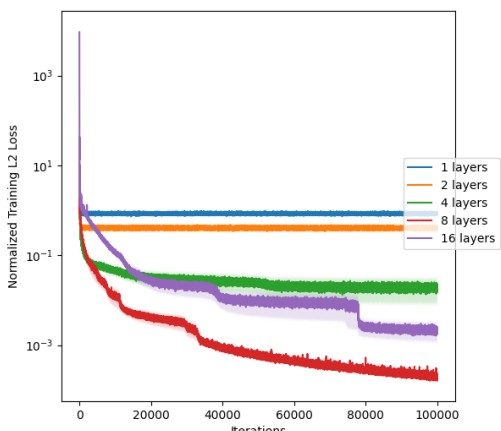

Figure 14: Real NVP Regressed on 4-D Linear Functions

Figure 15: Real NVP Regressed on 8-D Linear Functions

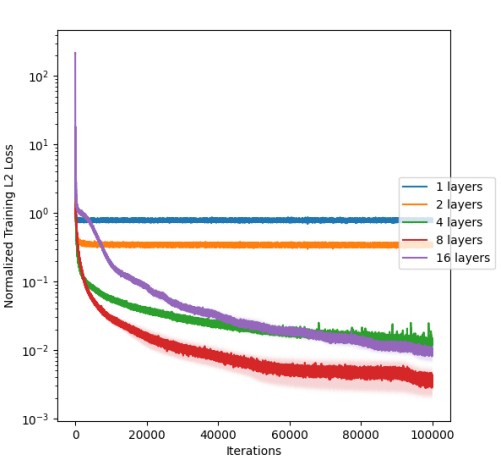

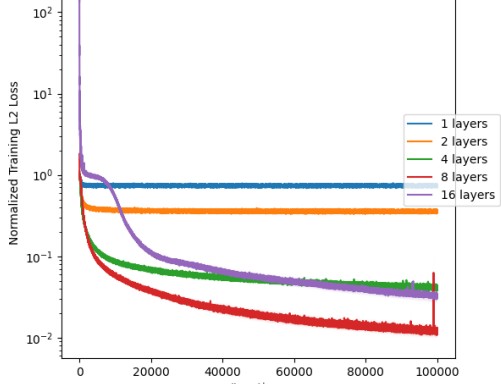

Figure 16: Real NVP Regressed on 16-D Linear Functions

Figure 17: Real NVP Regressed on 32-D Linear Functions

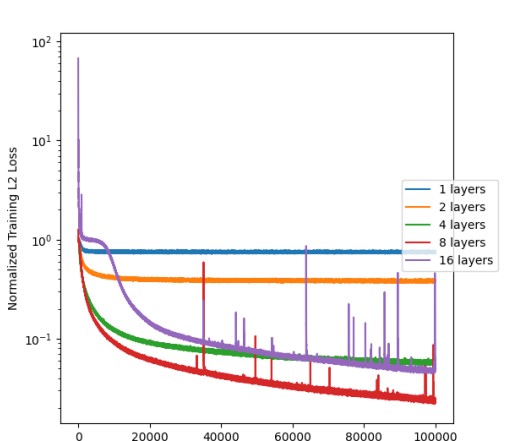

Figure 18: Real NVP Regressed on 64-D Linear Functions

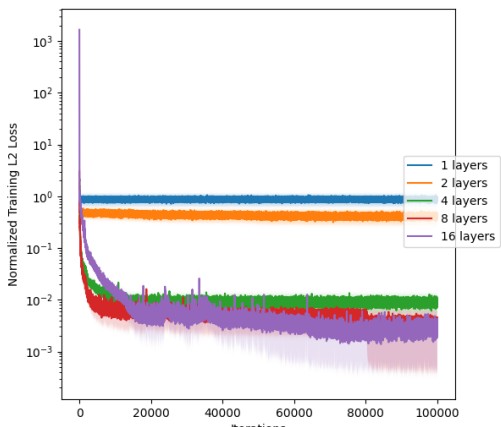

Figure 19: Real NVP Regressed on 4-D Toeplitz Functions

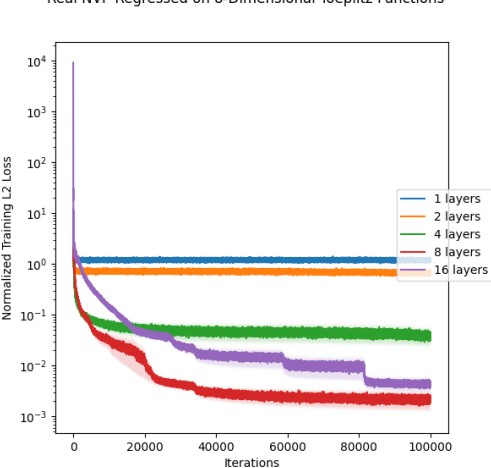

Figure 20: Real NVP Regressed on 8-D Toeplitz Functions

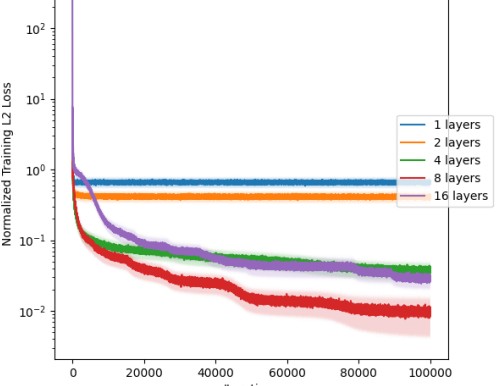

Figure 21: Real NVP Regressed on 16-D Toeplitz Functions

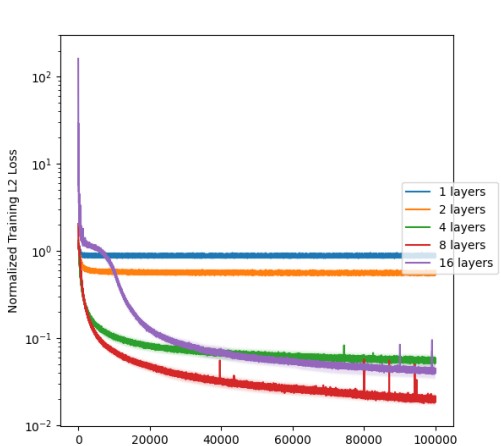

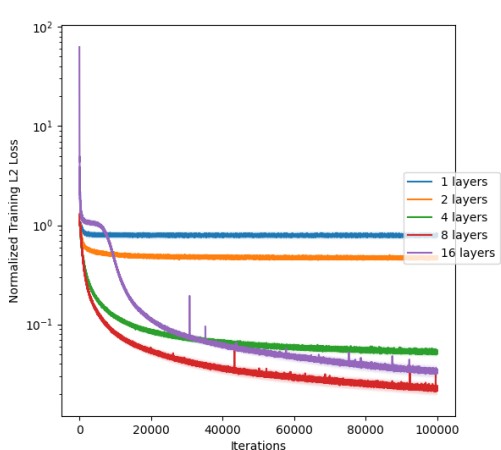

Figure 22: Real NVP Regressed on 32-D Toeplitz Functions

Figure 23: Real NVP Regressed on 164-D Toeplitz Functions

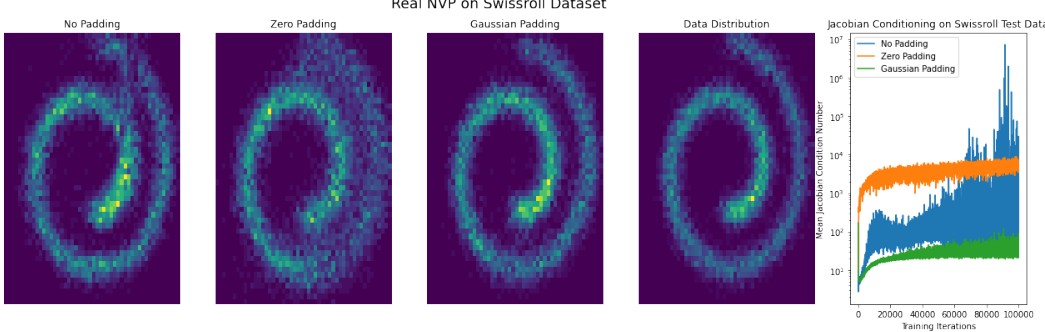

Figure 24: Real NVP on Swissroll Dataset

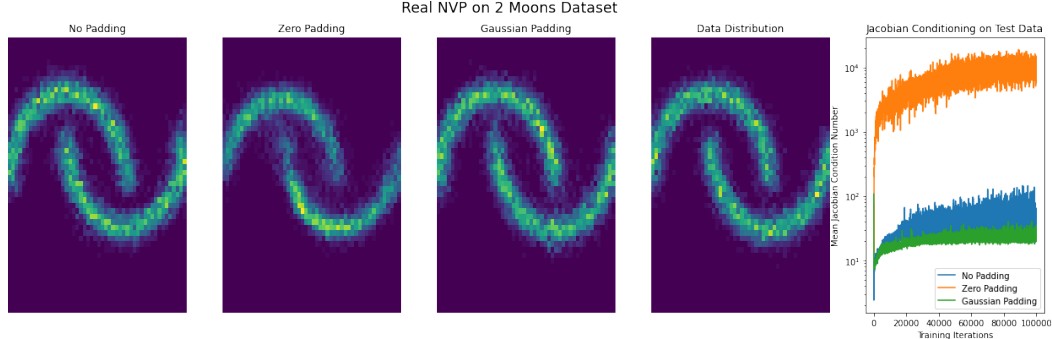

Figure 25: Real NVP on 2 Moons Dataset

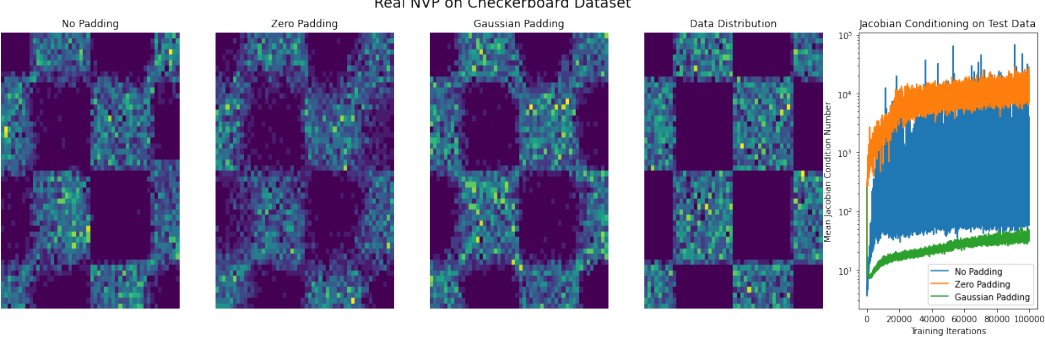

Figure 26: Real NVP on Checkerboard Dataset

