# OpenReview forum: "Representational aspects of depth and conditioning in normalizing flows"
_ICLR.cc/2021/Conference — Reject_

### Official Review · AnonReviewer3 · 2020-10-28
**review of Representational aspects of depth and conditioning in normalizing flows**

**Rating:** 6
**Confidence:** 3

**Review:**

**Summary**
This paper studies theoretical properties of flow models, mainly focussing on affine coupling layers. The authors investigate several questions on the representational capacity of flow models, focusing on the role of flow depth and the regularity/conditioning of the flow model.
Their findings can be summarized as follows: 1) The authors show that an invertible flow model (not necessarily of the affine coupling form) that needs to match the generated data of a non-invertible generator needs to be substantially deeper than the non-invertible generator, while maintaining roughly the same amount of parameters as the non-invertible generator. 2) The authors show that any linear invertible map can be learned with a constant number of affine coupling layers with a fixed partitioning. Therefore, approximations for learnable permutation layers such as the 1x1 convolutions of GLOW can be replaced by increasing the size of the flow network with a constant factor. 3) The authors show that if the Jacobian of affine flow models can be arbitrarily close to singular, affine flow models are universal approximators for distributions with bounded support. This results requires no zero padding of the input, as was the case of previous work [1]. 4) Finally, the authors explore the effect of zero padding or gaussian padding on 2d toy examples, and find that gaussian padding leads to better matching of the data distribution and a better condition number.


**Pros**
* The paper presents interesting fundamental questions on the representational capacity of normalizing flows, which can help practitioners in their design choices of normalizing flows.
* The authors clearly made an effort to try to make the proof sketches accessible. However, the proof sketches and the relation between theorems can be made much clearer. Improving along this axis would make the paper much more accessible to a wider audience.
* The authors empirically validate some of their theoretical results.

**Cons**
* There is quite some room for improvement on the side of the empirical results.
   - First, empirical results are obtained by optimizing with an objective that is not maximum likelihood but a regression objective. This makes the transferability of some of the results to flows trained with maximum likelihood (almost always the case) questionable. For instance, will the results on the different padding strategies in section 6.1 and appendix C2 still hold for maximum likelihood trained models? The argument for using the regression loss in 5.3 instead of maximum likelihood is also quite vague “...to minimize algorithmic (training) effects as the theorems are focusing on the representational aspects.” Does that mean that if you train with maximum likelihood you don’t obtain the same empirical results?
   - Second, the results for learning an invertible linear mapping with affine flow models is only done for a very particular construction of invertible linear mappings: the elements of the invertible matrix are sampled from a standard Gaussian. For this particular case it indeed seems that the practical number of affine coupling layers needed to approximate an invertible linear mapping is lower than the suggested upper bound (of 47). However, it is unclear if this observation extends to other invertible mappings that are not random Gaussian matrices.
   - Third, the empirical evaluation in section 5.3 could also be done with nonlinear affine coupling layers to show how this affects the practical results relative to the bounds that are claimed to also hold for this type of affine coupling layers.
   - Finally, the conclusion that Gaussian padding works better in practice than zero padding is based only on experiments with two-dimensional toy examples. In higher dimensional cases such as image datasets it is unclear if this still holds, as dimensionality can potentially play an important role here.

* The exposition and conditions for some of the theorems and how they relate to one another could be made clearer. For instance, how does the universal approximation without padding theorem (theorem 4) relate to the additional remark in section 5.3, where it is described that for a distribution generated by applying an elementwise tanh on a Gaussian random variable, this distribution cannot be modeled with a finite number of affine coupling layers? I appreciate that sections 4, 5, and 6 try to sketch the proofs of the theorems to give more insight to the reader. However, the proof sketches are not much clearer than the actual proofs in the appendices. I imagine some of the explanations could be made more accessible with visualizations. It would also be helpful for instance to relate theorem 1 and theorem 4 and their relationship.

Some of the details of the theorems are a bit confusing:
* In Theorem 2, why is only the case det(T)>0 considered? Could it also hold for invertible matrices with det(T)<0 such as a permutation with determinant equal to -1? Should in that case the diagonal matrices B and C have entries smaller than zero?
* I don’t understand the paragraph directly below theorem 3. Nonlinear affine coupling layers (where the nonlinearity refers to the nonlinearity of the scale and translation networks) are more flexible than linear affine coupling layers, so I would expect that theorem 2 can indeed also hold for nonlinear affine coupling layers, but that in theorem 3 the bound could be different because the nonlinear version is more flexible.
* The counting argument for the lower bound of K>=5 in theorem 3 is presented in a confusing way. In section 5.2, the number of parameters of a single coupling layer is indeed $d^2 + d$, so a product of $K$ coupling layers (either upper or lower triangular) is  $K(d^2 + d)$, and you would indeed expect 4 coupling layers to match the $4d^2$ parameters of the linear map. But the product terms in theorem 2 always consist of 2 coupling layers (one lower triangular $\in \mathcal{A_L}$ and one upper triangular coupling layer $\in \mathcal{A_U}$). So there the amount of parameters for a product of K terms would be $2K(d^2+d)$, leading to K=2 as an estimated sufficient amount based on parameter count alone, unless you assume that only one of the two coupling layers in each term is not the identity. I assume that the latter is the case as it would give you the freedom to represent the sequence of coupling layers as arbitrary combinations of upper and lower triangular coupling layers.
* In Theorem 1, if g is not invertible, how can we be sure that the distribution induced by the non-invertible mapping g is a valid distribution?

The cons currently outweigh the pros for me, leading to the rating of 5, but I do think this paper could be a valuable contribution to the normalizing flow community. Therefore, I hope the exposition can be made a little clearer during the revision period so I can raise my score.

**Minor comments/questions**
* Normalizing flows don’t need to start from Gaussian latent distributions as stated in the first paragraph of the introduction.
* Seems like $\sigma$ is used both for the activation function of layerwise invertible feedforward networks as well as the standard deviation for the gaussian distributions in theorem 1. These type of double usages make the paper less readable.
* Shouldn't the answer to question 1 be affirmative since you are confirming that such a distribution exists?
* in the paragraph below theorem 1, should $\Theta(k)$ and $\Theta(d^2)$ be $O(k)$ and $O(d^2)$?
* In definition 1, the scale and translation parameters of affine coupling layers are indicated with $g$ and $h$ respectively, but in some parts of the text, the authors refer to $s$ and $t$ for these factors (for instance just below theorem 3, and theorem 4). Please keep this consistent.
* Please include masked autoregressive flows in related work on generative normalizing flows [2].
* The related work in between the sections introducing the theorems and the proof sketches interrupts the flow of reading a little.
* Second paragraph page 6: typo “stanard” → “standard”.
* Just below lemma 6, should it be $\mathcal{A_L A_U A_L A_U}$ instead of $\mathcal{A_L A_R A_L A_R}$.
* In appendix C2, the data distribution of the swiss roll is depicted as the two moons data distribution.


[1] Huang et al. 2020, augmented normalizing flows: bridging the gap between generative flows and latent variable models. https://arxiv.org/abs/2002.07101
[2] Papmakarios et al., masked autoregressive flow for density estimation. https://arxiv.org/abs/1705.07057

---

> ### Author Response · Authors · 2020-11-15
> **Thank you for the thorough comments!**
>
> Thank you for your review and comments. We hope to use them to greatly improve the final paper. We appreciate your positive feedback and will address your concerns roughly in order.
>
> Re: empirical results -- to clarify, only our linear function experiments were trained with regression. There our theorem was strictly concerned with representational power results from Theorems 2 and 3. The regression objective is easier to optimize (since it’s supervised), and isolates checking representational power (as opposed to the influence of the extra layers on training dynamics for likelihood). Moreover, the likelihood objective is invariant up to a rotation when the function is linear, so it’s hard to measure the error of the learned network. In 6.1 we are in fact training by maximum likelihood. We’ve added a note to the paper clarifying this.
>
> We agree that the case of Gaussian matrices is a special case and we added a similar experiment with random Toeplitz matrices in Appendix C.1 in the updated version of our paper (Figures 9-13). Of course, whichever distribution of matrices we run experiments with -- the ultimate adjudication can only come by proving our bounds are tight (i.e. either finding a matrix which realizes the 47-matrix bound or improving the upper bound). Closing this gap is an interesting problem.
> We also added experiments which investigate whether adding nonlinearity to the affine couplings affects the number of layers needed, as suggested by the reviewer -- the resulting plots have been attached as a supplementary zip.
>
> Re: higher dimensional datasets -- we agree that it would be desirable to investigate the performance of padding on higher-dimensional datasets. Unfortunately, models which can produce high quality samples on image data so far are very large (see e.g. GLOW) which makes it difficult to perform full experiments at this scale. This is the reason we, like **many** other papers in this area, are running low-dimensional experiments.
>
> Re: relating the theorems -- we added some additional exposition relating the theorems. In particular, our comment on tanh is that it is not exactly representable but Theorem 4 shows it is approximately representable (with ill-conditioned models). If you have additional comments on what to clarify with respect to the relationships between the theorems we are happy to oblige!
>
> Re: det(T) > 0 --  the comment on Theorem 2 is correct, we only consider maps with det(T) > 0 as affine coupling models are orientation-preserving. If we were to add a diagonal layer with negative signs allowed, we would immediately recover the whole group.
>
> Re: the relationship between nonlinear models and Theorems 2 and 3 -- we note that though nonlinear models are more expressive than linear models, they both have Jacobians which are linear and of the form given. Therefore, the theorem still constrains what functions would be possible in the nonlinear setting.
>
> Re: inconsistencies in naming/counting-- Thank you for pointing out the inconsistencies in counting and naming matrices and coupling networks. We have corrected them in the paper. We also added the citation you noted.
>
> We hope we’ve been able to address your questions and concerns here, as well as with our updates to the paper.

---

> > ### Comment · AnonReviewer3 · 2020-11-23
> > **rebuttal response**
> >
> > Thank you for your rebuttal and for addressing some of my concerns. I have reread the revised submission. For completeness, I comment here on some of the points that were addressed or still leave some room for improvement:
> >
> > - my question/concern on the regression objective has been addressed.
> > - some clarifications have been added to the main text on how the theorems relate to each other.
> > - experiments with Toeplitz matrices are added as experiments to section 5.3. These are created by sampling constant values for the diagonals. The results show that in this case the Frobenius error on the parameters is larger than for the case of the independent Gaussian entries, hinting that the linear map with independent Gaussian entries is perhaps an easy example. These additional experiments are currently not mentioned in the main text, please mention them and adjust the conclusion accordingly.
> > - experiments with nonlinear flow models for learning linear mappings are added in the zip file of supplementary material. Please add these to the pdf directly of the supplementary material.
> > - on higher-D experiments:  Experiments on 2D examples and the experiments by Glow are two fairly extreme points in terms of how much compute they require. I am aware that compute resources are not equally distributed among researchers, but experimenting on higher-D image data, such as on a very small dataset such as MNIST, can be done with limited resources and more importantly they give valuable insight on the generality of the conclusions drawn in a paper. Indeed many papers show 2D examples, but these toy examples are often complimented with experiments on higher-D data such as images.
> >
> > One small thing I noticed when reading the revised paper: theorem 6 (restatement of theorem 3) mentions the condition $d>=4$. Theorem 3 does not.
> >
> > **Conclusion after reading the rebuttal and the revised submission**
> > I will raise my score to a 6 as the authors have made a clear effort to address some of the concerns, and because my answer to the question "Have I learned something from reading this paper?" is yes. The reason for not raising my score further is because it remains unclear how the results affect the learning of normalizing flows on more high-D data practically.

---

> > > ### Author Response · Authors · 2020-11-24
> > > **Thanks for your feedback!**
> > >
> > > Thank you for your continued feedback! We're glad the changes we made addressed your concerns -- we updated the submission to address this last round of changes (mentioning the additional experiments in the main text and adding the new experimental results to the appendix).

---

### Official Review · AnonReviewer2 · 2020-10-28
**This paper studies theoretically the effects of depth and conditioning in normalizing flows.**

**Rating:** 7
**Confidence:** 3

**Review:**

The normalizing flows (NF) are among popular generative models, as they have the capability of converting a simple base distribution to complex distributions by successively applying change of variable formula. More importantly, NFs let us do inference by maximizing exact likelihood functions instead of other approximate functions like ELBO in VAEs. NFs have the invertibility constraints which make the calculations of their Jacobean determinants computationally prohibitive and require some tricks that make them easier to compute. All these tricks, among them affine coupling layers are of great popularity, come at the price of losing expressiveness. Therefore, we might need more layers (deep) to compensate for that. However, we lack a theoretical understanding of how much deep is enough to guarantee best results.

This paper provides many useful and insightful theorems which sheds light on these kind of questions in using NFs. It specifically studies the affine coupling layers and provides an upper bound and a lower bound on the depth of the layers required to achieve a good performance. Then it investigates the effects of zero padding in the inputs of the NFs and shows that the reason that the NFs cannot work well with zero padding is actually related to poor conditionings of Jacobeans during the trainings.

Although this paper does not answers all the issues of the NFs, I think providing some interesting theorems on even simple questions, such as the effect of depth, could shed light on other questions in this field for other researchers. I have the following comments for this paper:

1- I think the formatting of this paper is wider than usual ones, which makes the reading difficult and it seems more compact. I think the authors should double check it to make it coherent with the other papers.

2- Although this is a theoretical paper, the number of experiments are limited. I believe that adding extra experiments can better reflect the value of the paper.

---

> ### Author Response · Authors · 2020-11-15
> **Thank you for the comments!**
>
> Thank you for the positive comments!
>
> Re: the formatting -- we apologize for the inconvenience -- we were allowed to fix the margins by the chairs so long as the length ends up < 9 pages. (Which it does.)
>
> Re: experiments: We did, as you noted, run some experiments verifying some of our theorems. If you have particular experiments to suggest that would help support the results and thrust of our paper, please let us know !

---

### Official Review · AnonReviewer4 · 2020-10-30
**The paper gives very thorough mathematical representation for two challenges related to normalization flows,but it is a bit unclear at which level the topic has been addressed in the previous research.**

**Rating:** 7
**Confidence:** 3

**Review:**

The paper gives very thorough mathematical representation for two challenges related to normalization flows, namely model’s large depth and conditioning which relates to the smallest singular value of the forward map. Topic is presented in a very orderly and comprehensive manner. All variables and concepts are explained and presentation is clear. Text and appendices give proofs for everything that has been discussed and appendices extensive presentation of experiments.

The cons of the paper are that it positions itself for previous research quite loosely. The paper would be a good handbook on the mathematics of the subject, but it is unclear at which level the topic has been addressed in the previous research, although the related work is presented in a short section. Also, it would be interesting to discuss how the obtained results could be considered when e.g. modelling the complex distributions.

---

> ### Author Response · Authors · 2020-11-15
> **Thank you for your comments!**
>
> Thank you for the positive comments!
>
> We have added a few more references to the related literature, which hopefully gives you a better sense of the position of our paper relative to them. Since space is limited, we kept our literature survey constrained to strictly relevant papers (a lot of which are mathematical in nature, since so is our paper). If there are some particular paper(s) you think we missed, we are happy to add them/discuss the relationship to our results.

---

### Official Review · AnonReviewer1 · 2020-11-06
**A potentially worthwhile topic, but questionable execution.**

**Rating:** 3
**Confidence:** 4

**Review:**

#### DESCRIPTION
This paper considers the representational ability of normalizing flows in terms of their overall size (depth, no. of parameters etc) and how they choose a partition for coupling layer transformations.

#### DISCUSSION

While I think exploring explicit limitations of the representational ability of normalizing flows imposed by the invertible constraints and bipartite partitioning of coupling layers, I'm not sure that (i) the specifics discussed in this paper get to the heart of the matter, and (ii) the points that *are* made are hard to understand and stand behind.

Specifically, I have the following main concerns:

"Empirically, these models seem to require a much larger size than other generative models (e.g. GANs) and most notably, a much larger depth." -- what does it mean for one generative model to be "bigger" than another, or to have "larger depth"? Is it number of parameters, or memory cost? The number of transformations in a flow and the number of layers in a GAN are not really comparable. You could also argue that normalizing flows could be adapted to have a memory cost constant in their depth, since their activations can be reconstructed on the fly for backprop.

"Question 1" -- the answer to this is yes, example: in R^2, the function f(x1, x2) = (x1, x1) (where x = (x1, x2) ~ N(O, I)) induces a distribution on the line x2 = x1 which cannot be represented by an invertible mapping. What you really mean to discuss is *to what extent* enforcing invertibility impacts representational ability, not *whether* enforcing invertibility impacts representational ability, because it does.

I am finding it very difficult to understand what Theorem 1 is saying, other than that it's in some sense trying to formalize the fact the invertible functions impose representational constraints. Specifically: What does it mean for a network to have size O(k)? What is k? It seems to be defined as k = o(exp(d))? What does that mean? What does it mean for the depth to be k / p = o(exp(d)) / no. of parameters? Why is the number of parameters per layer p the same for every layer? It seems to me as if the statements here are vague and not well-defined.

"Question 2" -- I don't understand the point of this question. Why is it worthwhile to have a result about how many affine coupling layers with a fixed partition are needed to compensate for the removal of a 1 x 1 convolution? From the Glow paper, the 1 x 1 convolution added increased the parameter count of their model by 0.2%, and the wallclock time for training increased by approximately 7%. Neither of these are prohibitive costs which would warrant careful examination of the necessity of the 1 x 1 convolution.

Like Theorem 1, I'm also not sure of the meaning/point of Theorem 2. In particular, "any applications of permutations to achieve a different partition of the inputs can in principle be represented as a composition of not-too-many affine coupling layers." How many is not-too-many? Even if we could be specific, why does it matter that we can compensate for the removal of 1 x 1 convolutions with more affine coupling layers?

Sections 4 & 5 & 6: I'm finding it very difficult to parse what's going on here. The sections are technically dense, draw on a wide range of formal results in passing, and it's hard for me to understand how the material is relevant or necessary. Moreover, why are there extensive proof 'sketches' in the main text at all? Where are the actual proofs?

The experimental results are toy, and I'm not entirely sure what they're trying to show. What exactly is happening with the padding? As in Huang et al (2020), are the authors now doing variational inference in an extended space? None of this is clear.

It would be remiss of me not to mention the fact that the paper has altered the margins of the ICLR template to increase the amount of material. For better or worse, an 8-page conference paper is the widely used format for machine learning publications. While I'm sympathetic that sometimes this can be an undue constraint, it provides a level playing field, and sharpening a message to fit within imposed limits is part of research. Deliberately altering the margins (i) makes the job of a reviewer more difficult because while reviewing they have to reason about which parts of the paper will be removed from the main text to fit in the final template, and (ii) removes the need for the authors to reflect on whether their message is focused enough. Content aside, the structure and layout of this paper could be significantly improved -- the abstract does not need to be multi-paragraph, the sweeping technical details are introduced quickly and poorly motivated, and I found the overall narrative confusing and difficult to follow.

#### EXTRA NOTES

"we can efficiently evaluate the likelihood of a data point" -- the likelihood is a function of the parameters, not data.

"which model distributions as pushforwards of a standard Gaussian" -- the base distribution in a normalizing flow is not necessarily Gaussian.

In section 2.2, computational constraints are not necessarily the reason for not using invertible weight matrices and invertible pointwise nonlinearities -- the O(d^3) determinant calculation can be sidestepped by parameterizing the weight matrix using e.g. an LU decomposition.

#### CONCLUSION

As I mentioned at the beginning, I think an examination of the representational limitations of normalizing flows in terms of their overall size and how they are affected by a chosen partitioning may be worthwhile, but I feel as if this paper doesn't adequately address these questions, and what it does say is difficult to understand.

---

> ### Author Response · Authors · 2020-11-15
> **Thank you for review -- please consider both expository and formal parts of our paper (1/2)**
>
> Thank you for your review and comments on the paper. We’d like to first clarify two general points, then we will answer detailed queries.
>
> First, we structured the paper such that there is informal discussion to motivate the results presented, a summary (sketches) of the proof techniques used, and a full presentation of the proofs in the appendix. It seems several of your questions are comments and concerns about our informal section, treating it as if it is formal mathematics. This feels very uncharitable: we do actually present precise theorem statements and proofs to bear that scrutiny, but the informal discussion is less precise so as to be approachable, especially by readers who are not specialists in this area.
>
> Second, in theoretical study of deep learning (and machine learning in general) it is common to split analysis into a trichotomy of questions about representational power (of a class of predictors, probabilistic models, etc.), statistical complexity (e.g. quantities like VC dimension, Rademacher complexity, etc), and algorithmic complexity (e.g. can training be done efficiently, does gradient descent get stuck in suboptimal local minima, etc.). We explicitly approach the first topic in this work. In works on representation it is typical to consider constraints to the class the model belongs to (e.g. disallowing 1x1 convolutions) and quantify how much bigger it would need to be made to account for this restriction. (There is a long tradition of this in theoretical computer science and machine learning, specifically deep learning, e.g. how much bigger a shallow network needs to be to approximate a deeper network.)
>
> We now proceed to answer individual questions.
>
> Re: size and depth --  in our paper, size is taken to mean the number of parameters (i.e. trainable weights) of a model. As we discuss in the abstract, the primary reason we care about *depth* is due to gradient vanishing / exploding (equivalently, the conditioning of the Jacobian, which in general will also be worse for a deeper network). From this aspect, it’s fair to compare the number of layers in a generator with the number of affine couplings in a flow. In fact, since in Theorem 1 we are lower bounding the depth of the normalizing flow model in terms of number of transformations, each of which may further be deep, we are in fact being conservative (the “effective depth” in terms of the Jacobian conditioning should be the number of transformations x depth of each of those transformations.  We provide some “back of the envelope” calculations in the paragraph after Theorem 1.) We don’t consider algorithmic questions such as the memory complexity of training here.
>
> Re: Question 1 -- the question is part of the exposition, not a formal statement (see our first general point). We added the phrase “approximated by” to avoid confusion with distributions with degenerate supports (this is of course, a trivial obstacle for invertible architectures -- though only if we are aiming to *exactly* represent the distribution). The question additionally asks for a *depth* separation: namely, we want a distribution that can be written as the pushforward through a small (but potentially non-invertible), shallow neural network, that cannot even be approximated unless a much deeper invertible architecture is used. The questions regarding $k$ seem to stem from the following confusion: $k$ is a parameter. Writing $k = \exp(o(d))$ just means "for every $k$ that isn’t too big (i.e. k grows slowly with the exponential of d asymptotically), the result in Theorem 1 holds". It isn't a definition of k. This is standard notation, but we added a bit more exposition in the paper to this effect. The size of our networks being $O(k)$ just means the number of parameters of the networks is $O(k)$. Finally, the number of parameters $p$ per transform is an upper bound, and can be taken to be $p = \max_i p_i$ for layers with $p_i$ parameters per layer.
>
> Re: Question 2 -- we note that having an additional type of layer gives another knob to tune when designing a network, so it is useful to know to what extent this knob is necessary. Our work is the first to show that these layers do not help much with respect to representational power (see our second general point about ways to theoretically study questions in deep learning). Your comment seems to be focusing on the increase in train-time (in terms of wall-clock time) by adding some amount of 1x1 convolutions. This does not get at the issue at all of how much we are "handicapping" the representational power of the model when disallowing these layers.
> We also give a precise number (24) for “not-too-many” in the statement of Theorem 2, additionally in the exposition just two lines under the “not-too-many” sentence. (See our first general point about expository vs formal parts of our paper.)

---

> > ### Author Response · Authors · 2020-11-15
> > **Thank you for review -- please consider both expository and formal parts of our paper (2/2)**
> >
> > Re: Sections 4, 5, and 6 -- we give proof sketches for our main results. Due to the length restrictions, it’s common in theory-leaning papers to defer the full proofs to the appendices, while conveying the main techniques in the first 8 pages of the paper. We note that there are copious references/hyperlinks, nearly after each lemma/theorem in these sections to the appropriate appendix where the full proof is presented.
> >
> > Re: experimental results and padding: we explain the setups in Sections 5.3 and 6.1, and again in Appendix C in greater detail. The experiments in Section 5.3 are intended to verify the tightness of Theorems 2 and 3 (namely, how many linear affine layers are necessary to simulate a random linear map). In Section 6.1, we verify (on low-dimensional synthetic data) that Gaussian padding (i.e. for every training data point, we add iid Gaussian samples in the extra dimensions) works better than 0-padding or no padding, and that the condition numbers of the Jacobians are better in the Gaussian padding case (thus, lending credence that the poor conditioning in our universal approximation theorem, as well as existent universal approximation results for zero padding are an issue in practice too, and not merely a theoretical artefact). We are not claiming anything about a relation to variational inference (or really anything deeper, we do not have any theoretical results about Gaussian padding), just empirically studying the condition number behavior under padding. We agree that it would be desirable to investigate the performance of padding on higher-dimensional datasets (as opposed to toy ones). Unfortunately, models which can produce high quality samples on image data so far are very large (see e.g. GLOW) which makes it difficult to perform full experiments at this scale. This is the reason we, like many other papers in this area, are running low-dimensional experiments.
> >
> > Likelihood: by likelihood of a data point, we mean the probability of generating this data point from a model with certain parameters. This is a function of both the data and parameters. The average likelihood (averaged over data) will be a function only of the parameters.
> >
> > Base distribution: You’re correct in saying that a normalizing flow can use other-than-Gaussian base distributions. We added a sentence to that effect.
> >
> > Computational costs: We believe that it is fair to note the computational cost of Jacobian determinant computation, as this is the original justification for e.g. the RealNVP architecture and the standard one for many kinds of normalizing flows (see, e.g. Section 3.2 in “Density Estimation Using RealNVP“). We agree this is not the only benefit of using affine coupling networks and we edited to emphasize this.
> >
> > Re: the formatting -- we apologize -- we were allowed by the chairs to fix the margins so long as the length ends up < 9 pages. (Which it does.) We assure you this was not intentional, and we apologize for the inconvenience.

---

> > > ### Comment · AnonReviewer1 · 2020-11-20
> > > **Response**
> > >
> > > "Re: Sections 4, 5, and 6 -- we give proof sketches for our main results."
> > >
> > > I thought the goal of a proof sketch was to 'sketch' the core arguments of a proof in a few lines, where the full proof would fill in the required details. The material in sections 4, 5, and 6 spans approx. 3 pages and I find it quite dense and difficult to follow. Why not just take the majority of these sketches and present them cohesively in a full proof?
> > >
> > > "The experiments in Section 5.3 are intended to verify the tightness of Theorems 2 and 3 (namely, how many linear affine layers are necessary to simulate a random linear map)."
> > >
> > > I presume by linear affine layers you mean affine coupling layers? As I've mentioned above, I don't really see what this tells us. You're using coupling layers as a regression tool to fit a linear map -- what does this tell us about the representational power of general invertible architectures on real problems? I can fit the map z -> Az where z is standard normal and A is invertible with a single invertible linear layer using e.g. the LU decomposition, and I'll recover a factorization of the covariance, which I can estimate directly by just taking the empirical covariance of the data in any case, since the random variable Az is actually Gaussian. What is this experiment trying to show?
> > >
> > > "In Section 6.1, we verify (on low-dimensional synthetic data) that Gaussian padding (i.e. for every training data point, we add iid Gaussian samples in the extra dimensions) works better than 0-padding or no padding, and that the condition numbers of the Jacobians are better in the Gaussian padding case (thus, lending credence that the poor conditioning in our universal approximation theorem, as well as existent universal approximation results for zero padding are an issue in practice too, and not merely a theoretical artefact). We are not claiming anything about a relation to variational inference (or really anything deeper, we do not have any theoretical results about Gaussian padding), just empirically studying the condition number behavior under padding"
> > >
> > > (i) Zero-padding of the input is a non-invertible transformation -- samples generated by the flow will not necessarily lie on the manifold defined by the zero-padding embedding operation. How are you accounting for this so that the inverse is constrained to always lie on the correct manifold?
> > > (ii) Gaussian padding of the input means you now have a joint distribution p(x, z) where the padding z is standard normal and x is data. If you're not using variational inference, how are you maximizing the marginal likelihood p(x) under the model? Or are you doing something else?

---

> > ### Comment · AnonReviewer1 · 2020-11-20
> > **Response**
> >
> > "the primary reason we care about depth is due to gradient vanishing / exploding (equivalently, the conditioning of the Jacobian, which in general will also be worse for a deeper network)."
> >
> > If the primary motivation for your theoretical analysis is gradient quality/Jacobian conditioning, why are there no experiments quantifying the effect of a vanishing gradient or a poorly conditioned Jacobian as a function of "depth"?
> >
> > On the other hand, the experiment included in section 5.3 is a toy regression task fitting pairs (z, Az) where z is standard normal, and A is almost surely invertible, meaning Az is also Gaussian with mean 0 and covariance S = A A^T. What does this toy linear regression task using invertible networks consisting of an increasing number of coupling layers tell us about the representational power of invertible architectures in general?
> >
> > "we note that having an additional type of layer gives another knob to tune when designing a network, so it is useful to know to what extent this knob is necessary"
> >
> > Whether to add 1 x 1 convolutions (or more generally an invertible linear layer in place of a permutation) is a binary choice -- there is no knob to tune. Ablations across many papers, the Glow paper among them, demonstrate increased performance when invertible linear layers are included, and the cost is a slight increase in computation and memory. I don't see why it's necessary to delve deeply into the exact representational ability of this architecture tweak.

---

> > > ### Author Response · Authors · 2020-11-20
> > > **Response to your second round of responses**
> > >
> > > As before, we start with two high level points first.
> > >
> > > First, we do ask you to please read the paper (both the expository and formal parts) **fully**. For instance, you ask: *“You're using coupling layers as a regression tool to fit a linear map -- what does this tell us about the representational power of general invertible architectures on real problems?”* The experiments and Theorem 2 + 3 address **affine couplings** -- not **general** invertible architectures. We make it clear that we have some results that apply to general invertible architectures, and some to affine couplings both in the abstract and in the beginning of section 2.2. In fact,  this is the title of subsection 2.2 ! -- “Results about Affine Couling Architectures”.
> > >
> > > Second, respectfully, you are taking a very reductionist view of what is a meaningful contribution and direction of study. For instance, you write: *“Ablations across many papers, the Glow paper among them, demonstrate increased performance when invertible linear layers are included, and the cost is a slight increase in computation and memory. I don't see why it's necessary to delve deeply into the exact representational ability of this architecture tweak.”*  This attitude of “empirically, many papers have demonstrated phenomenon X, so it isn’t necessary to delve deeply into a formal study” assumes that once a phenomenon has been empirically consistently demonstrated to hold, any further inquiry is unnecessary. Much of machine learning research however is concerned with mathematically showing **why** and **when** certain phenomena occur, in the hope of deriving insights that down the line can be used for improving the practice of machine learning.
> > >
> > > To your concrete points:
> > >
> > >
> > > Re: what is the goal of the experiments in Section 5.3 :
> > > The experiments are verifying exactly what Theorems 2+3 are trying to pin down (which is what we say at the beginning of the section): how many *linear affine coupling layers* with a **fixed partition** are necessary to approximate a linear map.  Namely, Theorem 2 provides an upper bound of 47, and Theorem 3 provides a lower bound of 4 -- in the experiments we are trying to check for a *random* linear map, whether the true number is closer to the upper or lower bound. It’s of course true that a **general** invertible architecture (e.g. one parametrized by the L and U matrices) can do so trivially easy, but these experiments, like Theorems 2+3, address **affine couplings**. This is even the title of subsection 2.2 -- “Results about Affine Coupling Architectures”. (See our first general point.)
> > >
> > > Re: 1x1 convolutions as a knob and prior work on ablations:
> > > Please see our second general point. Also, yes, 1x1 convolutions are a *binary* knob. So is the choice of activation functions (e.g. ReLU vs tanh), the inclusion/not inclusion of BatchNorm layers, etc. -- yet there are many papers written to try to quantify the **effect** of such choices formally and mathematically. (Incidentally, ReLU has empirically been more than demonstrated to be superior in most settings, and BatchNorm has more than been demonstrated to help in practice.)
> > >
> > > Re: section 4/5/6 and proof sketches:
> > > We are sorry you find the sections dense, but the proofs aren’t simple enough to be sketched in a few lines -- though they are not much longer. For instance, the proof sketches of theorems 2 and 3 are just a couple paragraphs each (Sections 5.1+5.2). The sketch of Theorem 1 is on pg 6 -- it describes the main lemmas in the proof, along with a high-level summary of how these proofs proceed. We can’t possibly take the entire appendix and present it as a “cohesive proof” in the main text  -- that would be 10+ pages of mathematics.
> > >
> > > Re: zero and Gaussian padding:
> > >
> > > Zero padding: Yes, we are aware zero-padding is not invertible! We even say so on the top of page 5 in our comment about the Jacobian! Prior results that prove universal approximation of affine-couplings (i.e. Huang et al) prove universal approximation **exactly if you pad with zeros**. Note universal approximation implies **approximation** -- not **exact** representation. Of course, if you have a distribution supported on a low-dimensional manifold you can’t hope to **exactly** represent it with an invertible model.
> > >
> > > Gaussian padding: We are simply doing maximum-likelihood training on the padded data, which means we are maximizing the joint likelihood p(x,z) = p(x)p(z), z~N(0,I). Then, we plot x from samples (x, z) taken from the fitted distribution. The Huang et al paper calls this the Augmented Maximum Likelihood Estimator (AMLE) -- there is no variational inference involved in our training or sampling process.
> > >
> > > Re: depth vs conditioning
> > > The fact that the deeper architectures in general are more liable to vanishing/exploding gradients is well known. For affine couplings in particular, Figure 10 in this paper https://openreview.net/pdf?id=BJlVeyHFwH shows that depth exacerbates conditioning.

---

> > > > ### Comment · AnonReviewer1 · 2020-11-22
> > > > **.**
> > > >
> > > > “You're using coupling layers as a regression tool to fit a linear map -- what does this tell us about the representational power of general invertible architectures on real problems?”
> > > > Let me be clear: what does (a) using affine coupling layers as a regression tool to fit a linear map tell me about (b) the representational ability of affine coupling layers when used as a component in a normalizing flow for a realistic density estimation problem? In my opinion the toy linear regression problem here is so detached from an actual real-world use case as to be meaningless. If the experiment is meant as an empirical verification for the theoretical result, then I feel as if the theoretical result can't be all that relevant to practitioners either, since otherwise a stronger empirical evaluation would have been used.
> > > >
> > > > "This attitude of “empirically, many papers have demonstrated phenomenon X, so it isn’t necessary to delve deeply into a formal study” assumes that once a phenomenon has been empirically consistently demonstrated to hold, any further inquiry is unnecessary. "
> > > > A permutation is a fixed invertible linear transformation which permutes basis vectors. If we parameterize a learnable invertible linear transformation, then that is a strict generalization of a permutation. For instance, if we take the PLU decomposition of an invertible matrix, then we can think of a permutation as setting LU as the identity and choosing a random fixed permutation. More generally, we can fix P, learn LU, and then the parameterization searches one of D! subspaces of invertible matrices, where D is the dimensionality of the data. We can also avail of other decompositions which can search larger subspaces. With this strict generalization, and if the overhead is small for the parameterization, again I ask what the need is to delve deeply into how many affine coupling layers are needed to compensate for an invertible linear layer.
> > > >
> > > > I still don't understand what's going on with the padding. "maximizing the joint likelihood p(x,z) = p(x)p(z), z~N(0,I)" - what? You don't have z, it's not observed. Are you just concatenating Gaussian white noise to the data and then fitting that distribution with a flow? If so, why?

---

> > > > > ### Author Response · Authors · 2020-11-23
> > > > > **. response**
> > > > >
> > > > > Thanks for your comments.
> > > > >
> > > > > Again, you bring up the point that adding 1x1 convolutions doesn’t cost too much extra in terms of compute, while providing some benefits -- so why dig deeper? We already addressed this, but let us try one more time: there are **many** phenomena of this type in deep learning -- e.g. mild overparameterization doesn’t cost much in terms of compute but tends to help a lot for training to avoid poor optima; adding batch-norm doesn’t cost too much in terms of compute but helps training a lot.  There’s a lot of recent and exciting work in theoretical machine learning to try and understand precisely **why**  this happens.
> > > > >
> > > > > Concretely to the first two paragraphs of questions: yes, ablation studies in e.g. the GLOW paper show they are beneficial to include and add a relatively small computational cost. Do you not think it’s an interesting question to explore how the 1x1 convolutions help? For instance, precisely how much modeling flexibility do they add (what we study)? Yes, obviously, a model that includes 1x1 convolutions has more modeling power, but *how much*? Our results directly address this question by quantifying how much larger a network that doesn’t include 1x1 convolutions needs to be in order to simulate one that does include them. The answer is between 5 times and 47 times larger; experiments show that for random linear maps our upper bound is probably loose, and the factor is closer to the lower bound. These kinds of quantitative questions of how much power an extra “knob” allows are bread and butter in theoretical studies of deep learning: e.g. depth [1], depth vs width [2], smoothness [3], activation function [4].
> > > > >
> > > > > To your third paragraph: yes, we’re concatenating (in our terminology, padding) Gaussian noise and fitting that augmented data with a flow. The small confusion in the prior answer is that we reused z in our previous response -- in the context of this response, it is fresh noise -- so your interpretation is correct. As to “why” you would think padding is sensible: there are multiple reasons; a) padding increases the dimension of the inputs -- hence also of the latents -- this would seemingly increase the modeling power of the model; b) zero padding is in fact proved in Huang et al to be sufficient for universal approximation (the intuition is that the extra coordinates are used as a “scratch pad” for the approximating model, which is not available when there are no extra coordinates to pad); c) Gaussian padding specifically makes intuitive sense because unlike zero padding which obviously results in a degenerate model (i.e. a rank-deficient Jacobian), it doesn’t seem like Gaussian padding has this problem. (We don’t have theory for this, but we at least empirically verified that in the low-dimensional settings we tried, this is indeed the case.)  We discuss this in admittedly concise language in section 6.1
> > > > >
> > > > > Again, thank you for your comments.
> > > > >
> > > > > [1] Benefits of depth in neural networks, Telgarsky, 2016
> > > > >
> > > > > [2] Depth-Width Trade-offs for ReLU Networks via Sharkovsky's Theorem, Vaggos Chatziafratis, Sai Ganesh Nagarajan, Ioannis Panageas, Xiao Wang, 2019
> > > > >
> > > > > [3] Error bounds for approximations with deep ReLU networks, Dmitry Yarotsky, 2016
> > > > >
> > > > > [4] Representational Power of ReLU Networks and Polynomial Kernels: Beyond Worst-Case Analysis,
> > > > > Frederic Koehler, Andrej Risteski, 2018

---

### Author Response · Authors · 2020-11-10
**Fixed Margins**

Per the PC instructions, we have just uploaded a revision which corrects the margins in our paper -- we accidentally used margins which were too big vertically and too small horizontally. There are no content changes in this revision. Thanks everyone for your reviews, we will respond to them soon.

---

### Public Comment · ~Felix_Draxler1 · 2020-11-17
**Technical questions regarding the proofs**

This paper addresses an important question, and I read it with great interest.
However, there are a some places where I'm unable to follow the argument.
It would bet great if the authors could comment on my questions.


### Regarding Theorem 1

Theorem 1 is split into several Lemmas, of which I have questions on two:

*Lemma 8* is given without proof, arguing it was immediate.
I am not sure if it is indeed so simple. Specifically, I do not see where the property of $f$ being invertible is needed in the proof. When this property is in fact not needed, the Lemma merely states a general limitation of Lipschitz-bounded networks and applies to invertible and feed-forward networks, such as $g$ if it were also Lipschitz-constrained, alike. This contrasts with your goal of showing an aspect where feed-forward networks are superior to invertible ones. Can you elaborate on this?

In *Lemma 9*, you prove a statement about $g: \mathbb R^{d+1} \to \mathbb R^d$ (one additional input dimension), whereas Theorem 1 refers to a mapping $g: \mathbb R^d \to \mathbb R^d$. Is Lemma 9 still applicable in the proof of Theorem 1?


### Regarding Theorem 4

The proof of Theorem 4 (Universality without padding) is not immediate to me: You do not make the choice of the functions $\phi_1$ and $\phi_2$ explicit.
Could it be that you uploaded an outdated version of the proof, still saying "(fixme: details)" in the text?

---

> ### Author Response · Authors · 2020-11-17
> **Thanks for the questions!**
>
> Hi Felix, thanks for your thorough reading of our paper! Great questions!
>
> We’ll address your points in order:
>
> (1) Regarding Lemma 8: You’re correct, Lemma 8 doesn’t require invertibility as it is a general counting argument. Re: how it affects the “message” of the theorem: it’s not quite right to say we are showing something merely about “Lipschitz-constrained” networks -- but for Lipschitz architectures that can be viewed as sequences of transformations *with at most $p$ parameters per layer* (invertible or not).
> In particular, e.g. for layerwise invertible feedforward networks, $p$ is by virtue of invertibility bounded by $d^2$ (as opposed to arbitrary Lipschitz nets, which can expand and shrink at will). Also, for this architecture, the non-invertible networks are a measure-0 set, so there is no difference between counting the total number vs invertible networks.
>
> (2) Regarding Lemma 9: Lemma 9 is correct as written, however, there was indeed a typo in Theorem 1, which we have fixed. The network $g$ should be $\mathbb{R}^{d+1} \to \mathbb{R}$ as you note. (This doesn’t really affect the message of the result though: the requirement on $g$ is only that it’s shallow and small, so the latent dimension is not important, so long as it’s small.)
>
> (3) Regarding Theorem 4: Oops! Good catch, the “fixme” is a leftover comment :-) ! The maps  $\varphi_1$ and $\varphi_2$ are the maps guaranteed to exist by Brenier’s theorem which transport the Gaussian input to the target distribution. We added some text in that section of the appendix (bottom of pg 20, start of pg 21) clarifying their definition.
>
> Thanks again, hope that helps!

---

### Decision · Program_Chairs · 2021-01-07
**Final Decision**

**Decision:**

Reject

**Comment:**

The paper studies three aspects of the representational capabilities of normalizing flows, with a particular focus on affine coupling layers. Normalizing flows are valuable generative-modelling tools, so advancing our understanding of their theoretical properties is an important research direction.

Reviewers #2 and #4 found the contribution of the paper significant without expressing major concerns, and so recommended acceptance.

Reviewer #3 reviewed the paper very thoroughly, and expressed some concerns mainly about the experimental evaluation. Most of their concerns were addressed in the rebuttal, so they recommended weak acceptance, recognizing the merits of the paper but also pointing out the potential for improvement.

Reviewer #1 was the most critical: they expressed major concerns regarding the significance of the contributions and the overall clarity of the exposition. Despite a long exchange between the reviewer and the authors, a consensus was not reached, so the concerns remain.

The discussion so far has led me to believe that there are potentially valuable theoretical contributions in the paper, however it's clear that there is significant room for improvement in getting the contributions across. Given the strong concerns expressed, the lack of consensus, and the clear potential for improvement, I'm unable to recommend acceptance of the paper in its current form. However, I do believe that the work has potential, and I hope that the discussion here will help improve the paper for a future submission.